# Hamiltonian Neural PDE Solvers through Functional Approximation

**Anthony Zhou**
Carnegie Mellon University
`ayz2@andrew.cmu.edu`

**Amir Barati Farimani**
Carnegie Mellon University
`barati@cmu.edu`

## Abstract

Designing neural networks within a Hamiltonian framework offers a principled way to ensure that conservation laws are respected in physical systems. While promising, these capabilities have been largely limited to discrete, analytically solvable systems. In contrast, many physical phenomena are governed by PDEs, which govern infinite-dimensional fields through Hamiltonian functionals and their functional derivatives. Building on prior work, we represent the Hamiltonian functional as a kernel integral parameterized by a neural field, enabling learnable function-to-scalar mappings and the use of automatic differentiation to calculate functional derivatives. This allows for an extension of Hamiltonian mechanics to neural PDE solvers by predicting a functional and learning in the gradient domain. We show that the resulting Hamiltonian Neural Solver (HNS) can be an effective surrogate model through improved stability and conserving energy-like quantities across 1D and 2D PDEs. This ability to respect conservation laws also allows HNS models to better generalize to longer time horizons or unseen initial conditions.

## 1   Introduction

Physics is remarkably complex; a set of governing laws can produce endlessly diverse and interesting phenomena. In an effort to unify governing laws, Hamiltonian mechanics describes physical phenomena under the Principle of Least Action, whereby the trajectory of a system evolves because it has locally the least action. Remarkably, the minimization of action can derive nearly all physical phenomena, from kinematics to quantum mechanics, suggesting an underlying, fundamental property of nature. Rather than describing physics through forces and masses, such as in Newtonian mechanics, using energy and symmetries allows for an elegant and unifying perspective of physical systems.

Despite the long history of Hamiltonian mechanics, most current numerical and neural PDE solvers operate within a Newtonian framework. This has been very successful; forces and masses are easy to intuitively understand, and both numerical and neural solvers have found important, practical uses. However, motivated by natural conservation laws and symmetries that arise from Hamiltonian mechanics, we seek to investigate neural PDE solvers that operate in this framework. We find that the Hamiltonian framework brings inductive biases that improve model performance, especially when conserving energy-like quantities.

Extending Hamiltonian mechanics to neural PDE solvers requires both theoretical and implementation insights. In particular, most PDEs are used to describe continuum systems (fluids, waves, elastic bodies, etc.); as such, infinite-dimensional Hamiltonian mechanics are needed. In this setup, the Hamiltonian is defined as a functional that maps from input fields to a scalar, often interpreted as the energy of the system. The variational or functional derivative of the Hamiltonian then contains dynamical information that evolves the system. This introduces interesting modeling challenges; in particular, the need to approximate infinite-dimensional to scalar mappings and for the approximation to have functional derivatives.

39th Conference on Neural Information Processing Systems (NeurIPS 2025).

In this paper, we consider models based on a learnable kernel integral, a framework that can provably approximate linear functionals and implicitly learn functional derivatives. Through automatic differentiation, functional derivatives can be obtained and optimized in a Hamiltonian framework to predict future PDE states. This allows the development of Hamiltonian Neural Solvers (HNS), which we evaluate on three PDE systems and find both high accuracy and generalization capabilities, which we hypothesize arise from inductive biases to conserve energy. Code and datasets for this work are released at `https://github.com/anthonyzhou-1/hamiltonian_pdes`.

## 2 Background

**Hamiltonian Mechanics**   Hamiltonian mechanics is usually introduced in the discrete setting, where there are a set of $n$ bodies, each with a position and momentum. Therefore, the state of the system can be described by $2n$ quantities, usually assembled into a position and momentum vector $\mathbf{q}, \mathbf{p} \in \mathbb{R}^n$. In the discrete case, the Hamiltonian $\{\mathcal{H}(\mathbf{q}, \mathbf{p}) : \mathbb{R}^{2n} \to \mathbb{R}\}$ is a function that takes two vectors and returns a scalar $\mathcal{H}$, usually interpreted as the energy or another conserved quantity. Evolving the system in time is done through Hamilton's equations, often expressed in terms of the symplectic matrix $J$ and state vector $\mathbf{u}$:

$$\underbrace{\frac{d\mathbf{q}}{dt} = \frac{\partial \mathcal{H}}{\partial \mathbf{p}}, \quad \frac{d\mathbf{p}}{dt} = -\frac{\partial \mathcal{H}}{\partial \mathbf{q}}}_{\text{Hamilton's Equations}} \qquad \underbrace{\frac{d\mathbf{u}}{dt} = \begin{bmatrix} d\mathbf{q}/dt \\ d\mathbf{p}/dt \end{bmatrix} = \begin{bmatrix} 0 & \mathbb{I} \\ -\mathbb{I} & 0 \end{bmatrix} \begin{bmatrix} \partial \mathcal{H}/\partial \mathbf{q} \\ \partial \mathcal{H}/\partial \mathbf{p} \end{bmatrix} = J\nabla_{\mathbf{u}}\mathcal{H}}_{\text{Symplectic Form}} \qquad (1)$$

Derivatives can be evaluated from vector calculus identities; for clarity we consider the state vector $\mathbf{u}$ with $2n$ components and write $\frac{d\mathbf{u}}{dt} = [\frac{du_1}{dt}, \dots, \frac{du_{2n}}{dt}]$, and $\nabla_{\mathbf{u}}\mathcal{H} = [\frac{\partial \mathcal{H}}{\partial u_1}, \dots, \frac{\partial \mathcal{H}}{\partial u_{2n}}]$.

**Infinite-Dimensional Systems**   Many PDEs describe continuum systems, such as waves, fluids, and elastic bodies, where the positions and momenta of discrete bodies are not well defined. Despite this change, continuum systems can still conserve energy and be viewed in a Hamiltonian framework. Firstly, finite-dimensional vectors $\mathbf{q}, \mathbf{p} \in \mathbb{R}^n$ become generalized to functions $u(x, t)$ defined on continuously varying coordinates $x \in \Omega$ and at a time $t$. The function $u \in \mathcal{F}(\Omega)$ describes the state of the system by assigning a scalar or vector for each coordinate (such as velocity, pressure, etc.). Secondly, the Hamiltonian is generalized from a function $\{\mathcal{H}(\mathbf{q}, \mathbf{p}) : \mathbb{R}^{2n} \to \mathbb{R}\}$ to a functional $\{\mathcal{H}[u] : \mathcal{F}(\Omega) \to \mathbb{R}\}$ that returns a scalar given an input function. Lastly, the symplectic matrix $J$ generalizes to a linear operator $\mathcal{J}$, and the gradient $\nabla_{\mathbf{u}}$ becomes the variational or functional derivative $\frac{\delta \mathcal{H}}{\delta u} \in \mathcal{F}(\Omega)$. This leads to Hamilton's equations for infinite-dimensional systems, where $\epsilon \in \mathcal{F}(\Omega)$ is an arbitrary test function [1]:

$$\underbrace{\frac{\partial u}{\partial t} = \mathcal{J}\frac{\delta \mathcal{H}}{\delta u}}_{\text{Hamilton's Equations}} \qquad \underbrace{\int_\Omega \frac{\delta \mathcal{H}}{\delta u}(x)\epsilon(x)dx = \lim_{h \to 0} \frac{\mathcal{H}[u + h\epsilon] - \mathcal{H}[u]}{h}}_{\text{Functional Derivative}} \qquad (2)$$

From this background we can see why neural networks are a natural choice for modeling discrete Hamiltonian systems; in this setting, the Hamiltonian is a function and its gradient is naturally calculated with automatic differentiation. In the infinite-dimensional setting, learning the Hamiltonian functional from data becomes less obvious.

## 3 Functional Approximation

**Theory**   The development of functional approximators [2–5] can be seen a result of the Riesz representation theorem, a fundamental result in functional analysis [6]. We give some intuition here, but provide a full proof of linear functional approximation in Appendix C.1. We start by observing that a functional can be equivalent to the inner product over suitable functions:

**Theorem 3.1** (Riesz representation theorem). *Let $H$ be a Hilbert space whose inner product $\langle x, y \rangle$ is defined. For every continuous linear functional $\varphi \in H^*$ there exists a unique function $f_\varphi \in H$, called the Riesz representation of $\varphi$, such that:*

$$\varphi[x] = \langle x, f_\varphi \rangle \quad \text{for all } x \in H. \qquad (3)$$

Immediately, we can see that the Hamiltonian $\mathcal{H}[u]$ can potentially be expressed as an inner product $\langle u, \kappa_\theta \rangle$ over the input function $u$ and a learnable function $\kappa_\theta$. This changes the problem of approximating a functional to the problem of approximating a function, which neural networks can readily achieve through universal approximation theorems [7, 8].

A common inner product for arbitrary functions $u, v$ in a Hilbert space $H$ involves an integral: $\langle u, v \rangle = \int_\Omega u(x)v(x)\mathrm{d}x$. One can check that this satisfies the basic properties of inner products and is also valid for vector-valued functions $u, v$. Based on this formulation, functionals can be parameterized by an *Integral Kernel Functional* (IKF):

$$\mathcal{H}_\theta[u] = \int_\Omega \kappa_\theta(x)u(x)\mathrm{d}x \tag{4}$$

In this form, we observe a connection between the integral kernel functional and the integral kernel operator that neural operators are built on [9]. Specifically, the integral kernel operator is given by:

$$\mathcal{K}_\theta(u)(x) = \int_\Omega \kappa_\theta(x, y)u(y)\mathrm{d}y \tag{5}$$

where a learnable operator $\mathcal{K}_\theta$ acting on an input function $u(x) \in \mathcal{U}$ produces an output function $\mathcal{K}(v)(x) \in \mathcal{V}$ for Banach spaces $\mathcal{U}, \mathcal{V}$; this expression is equivalent to the IKF when evaluated at a single point $x$. We can make a similar modification to neural operators where the kernel is allowed to be nonlinear by giving function values as input: $\kappa_\theta(x, y, u(x), u(y))$. In our setting, we can adopt a nonlinear kernel by allowing the kernel to change based on the input function: $\kappa_\theta(x, u(x))$.

One shortcoming of this framework is that the approximation of nonlinear functionals is not proven. Most Hamiltonians are nonlinear with respect to $u$, however, we empirically find that the proposed architecture can still approximate nonlinear functionals by using a nonlinear kernel $\kappa_\theta(x, u(x))$. There are some potential hypotheses for this. The Riesz representation theorem is a strong statement, implying that for linear functionals, there is a single, unique $\kappa_\theta \in H$ that can be used with any $u \in H$ to approximate $\mathcal{H}[u]$. In deep learning we are less constrained; we can potentially allow $\kappa_\theta(u)$ to depend on $u$ and even relax the uniqueness of $\kappa_\theta$ to still be able to approximate $\mathcal{H}[u]$. Furthermore, under assumptions of regularity and compact support, for arbitrary functionals $\mathcal{H}$ we can always show that a function $f \in H$ exists such that $\mathcal{H}[u] = \langle u, f \rangle$ by constructing $f = \frac{\mathcal{H}[u]}{||u||_2^2}u$, with the norm: $||u||_2 = \sqrt{\int_\Omega u(x)^2 dx}$.

**Implementation**  There are two main implementation choices: approximating the integral and parameterizing the kernel. The integral can be approximated by a Riemann sum:

$$\mathcal{H}_\theta[u] = \int_\Omega \kappa_\theta(x, u(x))u(x)dx \approx \sum_i \kappa_\theta(x_i, u(x_i))u(x_i)\mu_i\Delta x \tag{6}$$

with quadrature weights $\mu_i$. Interestingly, the full Riemann sum can be computed since the sum is only calculated once; in neural operator literature, approximating the integral with a full Riemann sum is usually too costly since the sum is evaluated for every query point. This results in approaches that truncate the sum [10–12] or represent it in Fourier space through the convolution theorem [13]. In our implementation, we evaluate different quadratures but find that the trapezoidal rule is enough to accurately estimate the integral.

To parameterize the kernel, we take the perspective from the Riesz representation theorem. In the linear case, the kernel is simply a function $\kappa_\theta(x) \in H$ that maps input coordinates to values in the codomain. This is exactly what a neural field is; therefore, we can inherit the substantial prior work on neural fields [14] to effectively parameterize $\kappa_\theta$.

One useful concept is the conditional neural field. In particular, the nonlinear kernel $\kappa_\theta(x, u(x))$ can be seen as a conditional neural field whereby the kernel output changes based on the input function $u(x)$. This concept can also be further extended to distinguish between local and global conditioning [15]; in neural field literature, the field can be conditioned on local information $u_i = u(x_i)$ or global information $\mathbf{u} = [u(x_0), \ldots, u(x_n)]$. Conditioning mechanisms can also be implemented in various ways. In the simplest case, the conditional information $u_i$ or $\mathbf{u}$ is concatenated to the input coordinate $x$ [16–18], however, we adopt a approach based on Feature-wise Linear Modulation (FiLM) [19].

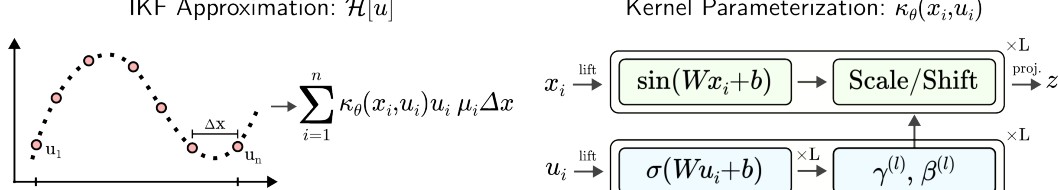

**Figure 1:** *Left:* The functional $\mathcal{H}[u]$ is approximated by an integral kernel functional (IKF). The integral is further approximated by a Riemann sum with a learnable kernel $\kappa_\theta$. *Right:* The kernel can be parameterized with various architectures and conditioning mechanisms. A local, FiLM-conditioned SIREN kernel is shown, where each SIREN layer has a dedicated FiLM network to produce scale and shift parameters. The output $z_i$ is for a single point $x_i$, which needs to be calculated $\forall i = \{1, \ldots, n\}$

Beyond conditioning mechanisms, an important consideration is the architecture of the kernel itself. A powerful neural field architecture is the sinusoidal representation network or SIREN [20]. SIRENs have shown good performance not only in fitting a neural field $\kappa_\theta$, but also in accurately representing gradients $\nabla \kappa_\theta$, which may help in fitting the functional derivative $\delta\mathcal{H}/\delta u$. With these architectural choices, a layer of the kernel $\kappa_\theta^{(l)}$ with local conditioning can be written as:

$$\kappa_\theta^{(l)}(x_i, u_i) = \gamma_\theta^{(l)}(u_i)\sin(Wx_i + b) + \beta_\theta^{(l)}(u_i) \tag{7}$$

where $\gamma_\theta^{(l)}$ and $\beta_\theta^{(l)}$ are the FiLM scale and shift networks at layer $l$. Each SIREN layer has its own FiLM network and each FiLM network can also have multiple layers. Inputs $x_i, u_i$ can optionally be lifted and the output $z_i$ can be projected back to the function dimension. Given the hierarchical nature of the proposed implementation, we also illustrate the architecture in Figure 1.

A final consideration during implementation is batching. The layer update in Equation 7 is written in pointwise form, but we can observe that the computation does not depend on other points $\{(x_j, u_j) : j \neq i\}$, and the weights are shared between points. Therefore, the forward and backward pass for each point can be done in parallel; in practice, the spatial dimension is reshaped into the batch dimension for efficient training and inference.

**Hamiltonian Neural Solvers**  So far, integral kernel functionals are largely agnostic to the downstream task, only requiring that the input function $u$ be discretized at a set of points $(x_i, u_i)$. To use functional approximators as a PDE solver, we introduce the Hamiltonian Neural Solver (HNS). The time evolution of PDEs can be described by Hamilton's equations, which depend on the operator $\mathcal{J}$ and the functional derivative $\delta\mathcal{H}/\delta u$. In Hamiltonian PDE systems, $\mathcal{J}$ is analytically derived or can be looked up, is guaranteed to be linear, and is usually simple. A common example in 1D is $\mathcal{J} = \partial_x$. Therefore, we choose to approximate $\mathcal{J}$ with a finite difference scheme, although it can also be learned with a neural operator. Lastly, we find that $\delta\mathcal{H}/\delta u$ can be computed directly with automatic differentiation. This is empirically shown in Section 4.1, but one way to see this is by considering the linear IKF. The gradient $\nabla_{\mathbf{u}} \sum_{i=1}^n \kappa_\theta(x_i)u_i \, \mu_i\Delta x = [\kappa_\theta(x_1), \ldots, \kappa_\theta(x_n)]$, up to a constant factor, which is the discretization of the learned function $\kappa_\theta(x)$. In this case, the functional derivative is approximated by an arbitrary function, which is the desired behavior.

Within this framework, the training is described in Algorithm 1. For Hamiltonian systems, we assume knowledge of the analytical form of $\mathcal{H}$ and $\delta\mathcal{H}/\delta u$, and labels for $\delta\mathcal{H}/\delta u$ or $d\mathbf{u}/dt$ can be calculated from training data using numerical schemes. The training loss can either be evaluated in the functional form $\delta\mathcal{H}_\theta/\delta u$ or the temporal form $\mathcal{J}(\delta\mathcal{H}_\theta/\delta u)$; we implement the former since it more directly optimizes $\kappa_\theta$.

---

**Algorithm 1** Training a HNS

1: **repeat**
2: $\quad \mathcal{H}_\theta \leftarrow \sum_{i=1}^n \kappa_\theta(x_i, u_i)u_i \, \mu_i\Delta x$
3: $\quad \frac{\delta\mathcal{H}_\theta}{\delta u} \leftarrow \text{autograd}(\mathcal{H}_\theta, \mathbf{u})$
4: $\quad \mathcal{L} = ||\frac{\delta\mathcal{H}_\theta}{\delta u} - \frac{\delta H}{\delta u}||^2 \text{ or } ||\mathcal{J}(\frac{\delta\mathcal{H}_\theta}{\delta u}) - \frac{d\mathbf{u}}{dt}||^2$
5: $\quad \theta \leftarrow \text{Update}(\theta, \nabla_\theta\mathcal{L})$
6: **until** converged

---

During inference, the current state $\mathbf{u}^t$ and coordinates $\mathbf{x}$ are used in a forward pass to compute $\mathcal{H}_\theta$. A backward pass is then used to calculate $\delta\mathcal{H}_\theta/\delta u$; using the operator $\mathcal{J}$, a prediction for $\frac{d\mathbf{u}}{dt}|_{t=t} = \mathcal{J}\left(\frac{\delta\mathcal{H}_\theta}{\delta u}\right)$ is calculated. The estimated derivative $\frac{d\mathbf{u}}{dt}$ is used to update the state $\mathbf{u}^{t+1} = \text{ODEint}(\mathbf{u}^t, \frac{d\mathbf{u}}{dt})$ using a numerical integrator. In our implementation, we use a 2nd-order Adams-Bashforth method.

| Metric | Base | | | OOD ($c_i \in [1,3]$) | | | Disc. ($x_i \in [-2,2]$) | | |
|---|---|---|---|---|---|---|---|---|---|
| | MLP | FNO | IKF | MLP | FNO | IKF | MLP | FNO | IKF |
| $\mathcal{F}_l[u]$ | 2.47e-5 | 2.76e-4 | 3.00e-16 | 0.046 | 0.268 | 2.13e-7 | 49.3 | 49.7 | 0.737 |
| $\delta\mathcal{F}_l/\delta u$ | 0.083 | 0.066 | 1.15e-3 | 0.114 | 0.122 | 1.15e-3 | 2.64 | 2.70 | 0.077 |
| $\mathcal{F}_{nl}[u]$ | 0.029 | 0.016 | 2.05e-3 | 6709 | 6493 | 2908 | 381 | 322 | 55.4 |
| $\delta\mathcal{F}_{nl}/\delta u$ | 1.33 | 2.10 | 0.045 | 1730 | 1723 | 1079 | 92.1 | 88.1 | 35.5 |

**Table 1:** Validation MSE is reported across different experiments with linear and nonlinear functionals. Errors are calculated in the scalar $\mathcal{F}[u]$ and gradient domains $\delta\mathcal{F}/\delta u$ to evaluate performance in fitting functionals and implicitly learning functional derivatives. Additionally, generalization to OOD inputs and unseen discretizations are evaluated. Parameter counts are: MLP (7.5K), FNO (10.9K) and IKF (4.3K); each experiment is repeated 6 times and errors are reported as the average across seeds.

# 4 Experiments

## 4.1 Toy Examples

To better understand and situate integral kernel functionals, we examine their ability to model simple, analytically constructed functionals and compare their performance to other architectures. The experimental setup is as follows: given a functional $\mathcal{F}[u]$, random polynomials $u(x) = c_0 x^p + c_1 x^{p-1} + \ldots + c_{p-1}x + c_p$ are generated by uniformly sampling $\{c_i \in [a,b] : i = 0, \ldots, p\}$. Therefore, the dataset consists of $N$ pairs of polynomials and evaluated functionals $(u^n(x), \mathcal{F}[u^n(x)])$ for $n = 1, \ldots, N$. In practice, each function $u^n(x)$ is discretized at a set of points $\{x_i : i = 1, \ldots, M\}$, such that it is represented as $\mathbf{u}^n = [u^n(x_1), \ldots, u^n(x_L)]$, and $\mathcal{F}[\mathbf{u}^n]$ remains a scalar.

We consider two cases: a linear functional $\mathcal{F}_l[u] = \int_{x_1}^{x_M} u(x) * x^2 \mathrm{d}x$ and a nonlinear functional $\mathcal{F}_{nl}[u] = \int_{x_1}^{x_M} (u(x))^3 \mathrm{d}x$. In both cases, constructed polynomials $u^n(x)$ can be substituted and the definite integral can be analytically solved to use as training labels. Additionally, the functional derivatives are $\frac{\delta\mathcal{F}_l}{\delta u} = x^2$ and $\frac{\delta\mathcal{F}_{nl}}{\delta u} = 3u^2$, which are used to evaluate the gradient performance of models. We restrict the degree of $u^n(x)$ to $p = 2$ and sample $c_i \in [-1,1]$; 100 training and 10 validation samples are generated for each case. Coordinates $x_i$ are uniformly spaced on the domain $[-1,1]$ with $M = 100$ points.

Two baselines are considered: (1) a **MLP** that takes concatenated inputs $[\mathbf{x}, \mathbf{u}] \in \mathbb{R}^{2M}$ and projects to an output $\mathcal{F}_\theta \in \mathbb{R}$ through hidden layers, and (2) a **FNO** with mean-pooling at the final layer to evaluate an operator baseline. The FNO baseline takes inputs with spatial and channel dimensions, or stacked inputs $[\mathbf{x}, \mathbf{u}] \in \mathbb{R}^{M \times 2}$ and projects to an output field $\mathbf{z} \in \mathbb{R}^M$, which is then pooled to $\mathcal{F}_\theta \in \mathbb{R}$. For simplicity, the integral kernel functional (**IKF**) is used with a local MLP kernel and is linear for $\mathcal{F}_l$ and nonlinear for $\mathcal{F}_{nl}$. In the nonlinear case, FiLM is not used and $[x_i, u_i] \in \mathbb{R}^2$ is concatenated as input to $\kappa_\theta$. Lastly, models are trained with the loss $\mathcal{L} = \sum_{n=1}^N ||\mathcal{F}[u^n(x)] - \mathcal{F}_\theta(\mathbf{u}^n, \mathbf{x})||_2^2$; training is supervised with the functional to evaluate if models can implicitly learn accurate derivatives.

To further evaluate generalization, after training we consider two cases: (1) **OOD**, where validation inputs $u(x)$ are sampled with polynomial coefficients $c_i \in [1,3]$ to model out-of-distribution functions, (2) **Disc.**, where validation inputs $u(x)$ are uniformly discretized on a larger domain $x_i \in [-2,2]$ with a larger $\Delta x$, while keeping $M = 100$. In each experiment, validation errors are reported with the metric $\frac{1}{N} \sum_{n=1}^N ||\mathcal{F}[u^n] - \mathcal{F}_\theta(\mathbf{u}^n, \mathbf{x})||_2^2$ or $\frac{1}{N} \sum_{n=1}^N ||\frac{\delta\mathcal{F}}{\delta u^n} - \mathrm{autograd}(F_\theta, \mathbf{u}^n)||_2^2$ to measure performance in fitting functionals and learning accurate functional derivatives.

Results are given in Table 1. We observe that architectures based on an integral kernel can accurately represent functionals and their derivatives, outperforming other architectures. Despite only training on scalars $\mathcal{F}[u]$, IKFs can implicitly learn the correct functional derivatives, as plotted in Figure 2. This is new; even on toy problems, traditional architectures do not have well-behaved functional derivatives, as such, learning in the gradient domain for Hamiltonian systems would not be possible. Although the gradient performance is poor, conventional architectures can still fit functionals and the MSE for validation samples within the training distribution (Base) is low. When considering generalization performance, IKFs can readily generalize to unseen input functions as well as unseen discretizations in the linear case. This empirically observed generalization is theoretically motivated; the Riesz

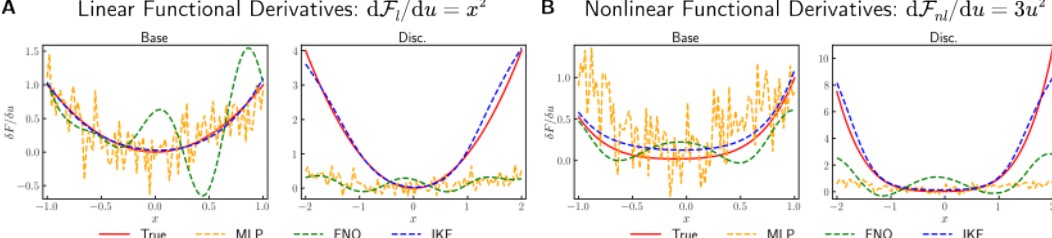

**Figure 2:** After training to regress $\mathcal{F}[u]$, the gradients of the network with respect to an input function $u$ are plotted for different models. *Left, A:* Neural functionals are able to accurately model derivatives, even on unseen discretizations, whereas other architectures fail even on their training distribution. *Right, B:* Nonlinear functionals have more complex derivatives that change with input functions $u$. In this instance, a quartic function is learned and the model extrapolates to unseen discretizations.

representation theorem guarantees that a single kernel $\kappa_\theta$ can represent a linear functional $\mathcal{F}_l$ for all possible input functions $u(x)$ (e.g., polynomial, exponential, etc.). Generalization performance with nonlinear functionals is more challenging, as larger polynomial coefficients $c_i$ and larger coordinates $x_i$ can quickly increase $F_{nl}[u(x)]$ due to large exponents. Despite this, IKFs still outperform other architectures in these domains, and learn accurate nonlinear functionals and their derivatives for samples within distribution.

### 4.2 1D Advection and KdV Equations

**Formulation** To evaluate the Hamiltonian Neural Solver (HNS) on PDE problems, we consider the 1D Advection and the 1D Korteweg–De Vries (KdV) equation. The Hamiltonians for each PDE are given as [21, 22]:

$$\mathcal{H}_{adv}[u] = \int_\Omega -\frac{1}{2}(u(x))^2 \mathrm{d}x, \qquad \mathcal{H}_{kdv}[u] = \int_\Omega -\frac{1}{6}(u(x))^3 - u(x)\frac{\partial^2 u}{\partial x^2}(x)\mathrm{d}x, \qquad (8)$$

Given an initial state $u^0$, the Hamiltonian for each PDE is conserved over time (i.e., $\mathcal{H}[u^t] = \mathcal{H}[u^0]$). For both PDEs, the operator $\mathcal{J}$ is defined as $\partial_x$. We can check that Hamilton's equations hold for the 1D Advection equation:

$$\frac{\delta\mathcal{H}_{adv}}{\delta u} = -u(x), \qquad \mathcal{J}(\frac{\delta\mathcal{H}_{adv}}{\delta u}) = -\frac{\partial u}{\partial x} = \frac{\mathrm{d}u}{\mathrm{d}t} \qquad (9)$$

which recovers the correct PDE for 1D Advection. A similar calculation can be done to recover the PDE for the 1D KdV equation ($u_t + uu_x + u_{xxx} = 0$) from Hamilton's equations by using $\frac{\delta\mathcal{H}_{kdv}}{\delta u} = -\frac{1}{2}u^2 - u_{xx}$. Therefore, given a training dataset of spatiotemporal PDE data, the labels $\frac{\delta\mathcal{H}}{\delta u}$ can be computed using finite differences at every timestep $t$. The gradients of the HNS network $\nabla_{\mathbf{u}}(\mathcal{H}_\theta(\mathbf{x}, \mathbf{u}^t))$ are then fitted to these labels.

**Evaluation** Data for the 1D Advection and KdV equations are generated from a numerical solver using random initial conditions [23, 24]:

$$u^0(x) = \sum_{i=1}^{J} A_i * \sin(\frac{2\pi l_i * x}{L} + \phi_i) \qquad (10)$$

where $L$ is the length of the domain, $A_i \in [-0.5, 0.5]$, $l_i \in \{1, 2, 3\}$, and $\phi_i \in [0, 2\pi]$; $J = 5$ for Advection and $J = 10$ for KdV. The Advection equation is solved on a uniform domain $x_i \in [0, 16]$ with $n_x = 128$ and the KdV equation is solved on a domain $x_i \in [0, 100]$ with $n_x = 256$. Both equations have periodic boundary conditions. For the Advection equation, 1024/256 samples are generated for train/validation, and for the KdV equation, 2048/256 samples are generated. To evaluate temporal extrapolation, the training dataset is truncated to a shorter time horizon. For the Advection equation, it is solved from $t = 0s$ to $t = 4s$ during training ($n_t = 200$) and evaluated from $t = 0s$ to $t = 20s$ during validation ($n_t = 1000$); a similar scenario is constructed for the KdV equation with $t = 0s$ to $t = 25s$ for training ($n_t = 50$) and $t = 0s$ to $t = 100s$ for validation ($n_t = 200$).

| Metric: | Adv
Roll. Err. ↓ | KdV
Corr. Time ↑ |
|---|---|---|
| FNO | $0.83_{\pm0.17}$ | $68.0_{\pm10.5}$ |
| Unet | $0.40_{\pm0.29}$ | $134.0_{\pm10.6}$ |
| FNO($\frac{d\mathbf{u}}{dt}$) | $0.048_{\pm0.007}$ | $77.6_{\pm3.6}$ |
| Unet($\frac{d\mathbf{u}}{dt}$) | $0.057_{\pm0.024}$ | $113.3_{\pm15.0}$ |
| HNS | $0.0039_{\pm0.0008}$ | $151.1_{\pm3.0}$ |

**Table 2:** Results for 1D PDEs. Parameter counts are: FNO (65K), Unet (65K), HNS (32K) for Adv, and FNO (135K), Unet (146K), HNS (87K) for KdV.

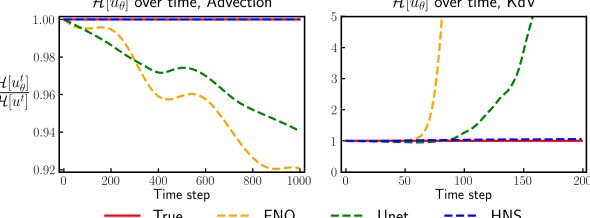

**Figure 3:** Hamiltonians for Adv and KdV for predicted trajectories, plotted over time. FNO and Unet models are shown with their $d\mathbf{u}/dt$ variants. HNS can better conserve Hamiltonians and remain stable over time.

Two baselines are considered, a FNO [13] and a Unet [25] model. Additionally, the HNS model is used with a nonlinear, FiLM-conditioned SIREN kernel. Global conditioning is used in the KdV experiments by calculating the scale/shift parameters as $\gamma_i^{(l)} = [\gamma_\theta^{(l)}(\mathbf{u}^t)]_i$ and $\beta_i^{(l)} = [\beta_\theta^{(l)}(\mathbf{u}^t)]_i$; the global FiLM networks are each parameterized by a shallow, 1D CNN. We find that global conditioning is needed in more complex PDEs since the kernel output $z_i = k_\theta(\cdot)$ can depend on more than just $u_i$ at a given point $x_i$. In particular, the Hamiltonian for the KdV equation has a $u_{xx}$ term that requires non-local knowledge of $\mathbf{u}^t$. Lastly, to ensure a fair comparison, we also train variants of the FNO and Unet baselines that predict temporal derivatives $d\mathbf{u}/dt$ [26], as opposed to the usual variants that predict $\mathbf{u}^{t+1}$; these use the same numerical integrator during inference as the HNS model.

The metrics that are reported are the rollout error $= \frac{1}{T}\sum_{t=1}^{T}\mathcal{L}(\mathbf{u}^t, \mathbf{u}_\theta^t)$, and the correlation time $= \max(t)\ s.t.\ \mathcal{C}(\mathbf{u}^t, \mathbf{u}_\theta^t) > 0.8$. The loss used is a relative L2 loss, or $\mathcal{L}(\mathbf{u}^t, \mathbf{u}_\theta^t) = \frac{||\mathbf{u}^t - \mathbf{u}_\theta^t||_2^2}{||\mathbf{u}^t||_2^2}$ and the correlation criterion $\mathcal{C}$ is the Pearson correlation. Rollout error is used to measure the accuracy of the predicted trajectory; for chaotic dynamics, rollout error is often skewed after autoregressive drift, therefore correlation time measures the portion of the trajectory that is accurate. Each metric is averaged over the validation set and the mean and standard deviation over six seeds is reported in Table 2. Furthermore, given a predicted trajectory $[\mathbf{u}_\theta^1, \ldots, \mathbf{u}_\theta^T]$ from a model, we can examine its Hamiltonian at each timestep and plot the scalar value $\mathcal{H}[\mathbf{u}_\theta^t]$ over time, shown in Figure 3.

We observe that the proposed HNS model is able to outperform current baselines, both on quantitative metrics and when conserving Hamiltonians over time. HNS is also efficient, capable of achieving better performance with nearly half the parameters of baseline models due to sharing weights between input points $x_i, u_i$. Furthermore, despite training on a shorter time horizon, HNS can extrapolate and remain stable in longer prediction horizons. This is more pronounced in the KdV equation, where initial states are smoother than later, more chaotic states. An example trajectory is shown in Figure 5, where the baselines struggle to predict later states since these are not seen during training.

### 4.3 2D Shallow Water Equations

**Formulation**   To evaluate HNS on a more complex system, we consider the 2D Shallow Water Equations (SWE). These equations are an approximation of the Navier-Stokes equations when the horizontal length scale is much larger than the vertical length scale. We consider its conservative form, which is often used as a simple model for atmosphere or ocean dynamics.

For vector-valued functions $\mathbf{u}(x, y) \in \mathbb{R}^d$, the operator $\mathcal{J}$ becomes a $d \times d$ matrix of operators. For the 2D Shallow Water Equations, the operator matrix $\mathcal{J}$ is given by [27–29]:

$$\mathcal{J} = \begin{bmatrix} 0 & -q & \partial_x \\ q & 0 & \partial_y \\ \partial_x & \partial_y & 0 \end{bmatrix}, \qquad q = \frac{\partial v_x}{\partial y} - \frac{\partial v_y}{\partial x} \tag{11}$$

where $q$ is the vorticity. The Hamiltonian for this system is:

$$\mathcal{H}[\mathbf{u}] = \mathcal{H}[v_x, v_y, h] = \int_\Omega \frac{1}{2}h(v_x^2 + v_y^2) + \frac{1}{2}gh^2 \mathrm{d}A \tag{12}$$

where $g$ is the gravitational constant and the velocities and height $v_x, v_y, h$ are usually defined on a grid (i.e., $v_x = v_x(x_i, y_j)$). One can verify Hamilton's equations and that $\mathcal{J}\frac{\delta\mathcal{H}}{\delta\mathbf{u}}$ recovers the 2D

| Model | Sines | Pulse |
|---|---|---|
| Transolver | $0.084_{\pm 0.0081}$ | $0.122_{\pm 0.0074}$ |
| FNO | $0.057_{\pm 0.002}$ | $0.117_{\pm 0.0009}$ |
| PINO | $0.053_{\pm 0.005}$ | $0.114_{\pm 0.0003}$ |
| Unet | $0.010_{\pm 0.0014}$ | $0.042_{\pm 0.0006}$ |
| HNS | $0.026_{\pm 0.0003}$ | $0.021_{\pm 0.0015}$ |

**Table 3:** Rollout errors for 2D SWE. Parameter counts are: FNO/PINO (7M), Transolver (4M), Unet (3M), HNS (3M). Models are evaluated on in-distribution ICs (Sines) or out-of-distribution ICs (Pulse).

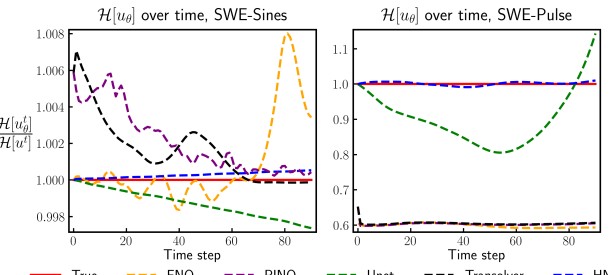

**Figure 4:** Hamiltonians for Sines and Pulse ICs, plotted over time. Despite potentially larger rollout errors, HNS can better conserve the Hamiltonian of predicted solutions.

Shallow Water Equations:

$$\partial_t h + \nabla \cdot (\mathbf{v}h) = 0, \qquad \partial_t \mathbf{v} + \mathbf{v} \cdot \nabla \mathbf{v} = -g\nabla h \qquad (13)$$

**Evaluation** Data for the 2D SWE system is generated from PyClaw [30–32] using random sinusoidal initial conditions (Sines) for the height:

$$h^0(x, y) = H + \sum_{i=1}^{J} A_i * \sin(\frac{2\pi m_i * x}{L_x} + \frac{2\pi n_i * y}{L_y} + \phi_i) \qquad (14)$$

with initial velocities set to zero. Constants are randomly sampled from $A_i \in [-0.1, 0.1]$, $m_i, n_i \in \{1, 2, 3\}$, and $\phi_i \in [0, 2\pi]$ with $J = 5$, $H = 1$, and $L_x = L_y = 2$. The domain is $(x_i, y_i) \in [-1, 1]^2$ with $n_x = n_y = 128$; each sample has $n_t = 100$ timesteps from $t = 0s$ to $t = 2s$. The gravitational constant is set to $g = 1$; 256 samples are generated for training and 64 samples are generated for validation. Periodic boundary conditions are used. To evaluate model generalization, we generate an additional test set of 64 samples with a random Gaussian pulse as initial conditions (Pulse). The initial height is set to an isotropic Gaussian centered at the origin with random covariance $\sigma \in [0.1, 0.5]$ and the height is scaled to be between $[1, 1.5]$; initial velocities are zero. The domain, BCs, and discretizations remain the same.

We consider benchmarking against Unet and FNO models, as well as PINO [33] and Transolver [34] models to compare against a physics-informed method and a newer attention-based surrogate. We instantiate the HNS model with a global, FiLM-conditioned SIREN kernel. Validation rollout errors for in-distribution ICs (Sines) and out-of-distribution ICs (Pulse) are given in Table 3, where means and standard deviations are calculated over three seeds. Additionally, the Hamiltonian over time for model rollouts are plotted in Figure 4. The derivative variants of baselines did not train stably, possibly due to sharp gradients that develop in the absence of viscosity, and are not reported. Additionally, the baselines are trained with the benefit of normalization, as the height channel has a different scale than the velocity channels; however, since the Hamiltonian is not invariant under scaling and shifting, normalization was not performed for the HNS model.

For test samples within the training distribution (Sines), conventional neural surrogates are able to perform well and achieve low loss, and HNS is on-par with these models. Adding a physics-informed loss in PINO only marginally increases the performance of FNO, which is consistent with its interpretation as a soft regularizer. When looking at the Hamiltonian of predicted solutions, we find that HNS can predict solutions that better conserve physical invariances. We validate this in Appendix B.1 by calculating the error of the Hamiltonian of predicted trajectories across different experiments. This ability to conserve the Hamiltonian helps HNS to generalize to unseen initial conditions, and it achieves better performance on the Gaussian Pulse test set. Interestingly, it also conserves the Hamiltonian on out-of-distribution inputs, suggesting a strong inductive bias in the model. To understand what mechanisms enable energy conservation, we also ablate different methods within HNS to understand their contributions, shown in Appendix B.1. Lastly, to visualize model performance, predicted trajectories of Sines and Pulse ICs are shown in Figures 6 and 7.

# 5 Related Works

**Hamiltonian Neural Networks**   Original work on Hamiltonian Neural Networks (HNNs) [35, 36] established them as a promising architecture to respect physical laws, while later work improved their performance [37–41], investigated the mechanisms behind HNNs [42], or used HNNs in different applications [43–45]. A closely related set of works develop neural networks in a Lagrangian framework [46, 47], which also preserve learned energies. More broadly, learning an energy and optimizing its gradient is an established framework in ML-based molecular dynamics simulation [48–52]. Despite research in this area, no prior works have considered learning with infinite-dimensional Hamiltonian systems and most HNNs have only been demonstrated in simple, particle-based systems.

**Functional Approximation**   While uncommon, there are prior works that investigate learning infinite-dimensional to scalar/vector maps. Most methods are based on a kernel integral and are used to fit functionals [2], construct functional autoencoders [4], model parameter-to-observable maps [3], or build infinite-dimensional discriminators for generative modeling [5]. The use of a kernel integral to model a functional is also well studied in machine learning for DFT, since the Hohenberg-Kohn Theorems guarantee that the total energy is the integral of an unknown function of electron density [53–55]. Lastly, a separate approach considers modeling functionals through the cylindrical approximation, rather than a kernel integral [56].

**Neural PDE Solvers**   Neural PDE solvers are a growing field, with many works proposing architectures to improve model accuracy [57, 34, 58], generalizability [59–61], or stability [62–65]. Specific to our work, many PDE solvers have also employed physics-based prior knowledge to improve model performance and adherence to governing laws. The most straightforward approach is to use physics-informed losses [66, 33, 67], but another, more fundamental approach is to enforce physical symmetries and equivariances in the network itself [68–72]. Interestingly, by Noether's theorem, every symmetry has an associated conservation law, which HNS relies on as an inductive bias; through this lens the current work is another way of softly enforcing symmetries in predictions.

# 6 Conclusion

**Limitations**   Despite an interesting theoretical basis and its novelty, Hamiltonian Neural Solvers have limitations that we hope future work can address. Firstly, the theory for nonlinear functionals is underdeveloped. There is also additional overhead during inference to backpropagate through the network and evaluate the linear operator $\mathcal{J}$. Scalability is also a concern due to the reliance on numerical quadratures. This increase in runtime is quantified in Appendix B.2

Additionally, implementing $\mathcal{J}$ without a neural operator may require care to avoid numerical errors; this is further discussed in Appendix D.1. Furthermore, HNS is only applicable to Hamiltonian systems; this would exclude systems with dissipation, although modifications exist to extend Hamiltonian mechanics to non-conservative systems [73–76]. Even for conservative PDEs, implementing a Hamiltonian structure is not as straightforward as training a neural solver with next-step prediction. Lastly, while not a limitation, an interesting observation is that introducing a loss on $\mathcal{H}[\mathbf{u}^t] - \mathcal{H}[\mathbf{u}_\theta^t]$ harms HNS performance; we hypothesize that since the Hamiltonian is conserved over time, learning the identity mapping is the easiest way to minimize this error and thus degrades model training.

**Outlook**   In this work, we have proposed a novel method for designing neural PDE solvers that respect conservation laws. We verify that kernel integrals are able to implicitly learn functional derivatives, as well as propose parametrization using neural fields. Using the resulting architecture in a Hamiltonian framework allows stable and energy-conserving predictions of 1D and 2D PDEs. These capabilities enable Hamiltonian Neural Solvers to generalize to certain out-of-distribution inputs, such as a longer time horizon or unseen initial conditions.

While learning functions and operators are dominant paradigms, approximating functionals may also have interesting uses. Functionals provide concise descriptions of physical systems by integrating physical variables, such as describing the energy of a molecule or the drag of an airfoil. Although concise, functional derivatives are often more relevant and can contain dynamical information or perhaps be used for optimization. We hope that future work can continue to improve functional approximators as well as take advantage of these models in new and insightful applications.

## Acknowledgments and Disclosure of Funding

We would like to thank Dr. Amit Acharya for his insightful discussions and help in conceptualizing this work.

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

# A Additional Visualizations

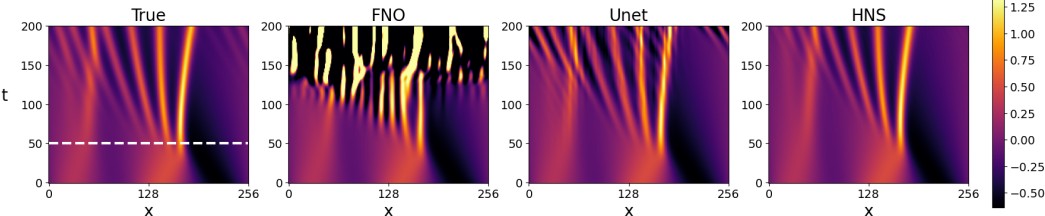

**Figure 5:** Predicted trajectories of the KdV equation. The white dashed line denotes the training horizon, where only states from $t = 0$ to $t = 50$ are seen during training. Despite not seeing later, more chaotic states, HNS still extrapolates beyond the training horizon by conserving the Hamiltonian, while other models struggle. FNO and Unet models are shown with their $d\mathbf{u}/dt$ variants.

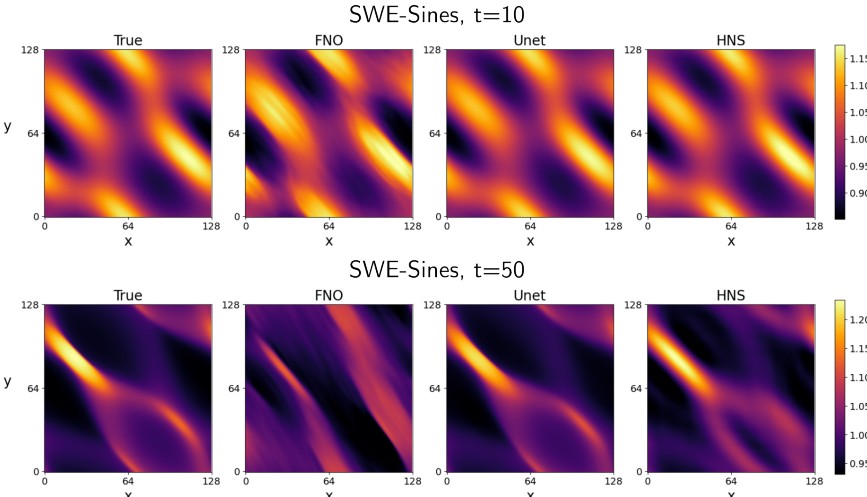

**Figure 6:** Predicted trajectories of the Shallow Water Equations (SWE) with sinusoidal ICs, displayed at $t = 10$ and $t = 50$. All models are able to achieve good performance on in-distribution samples.

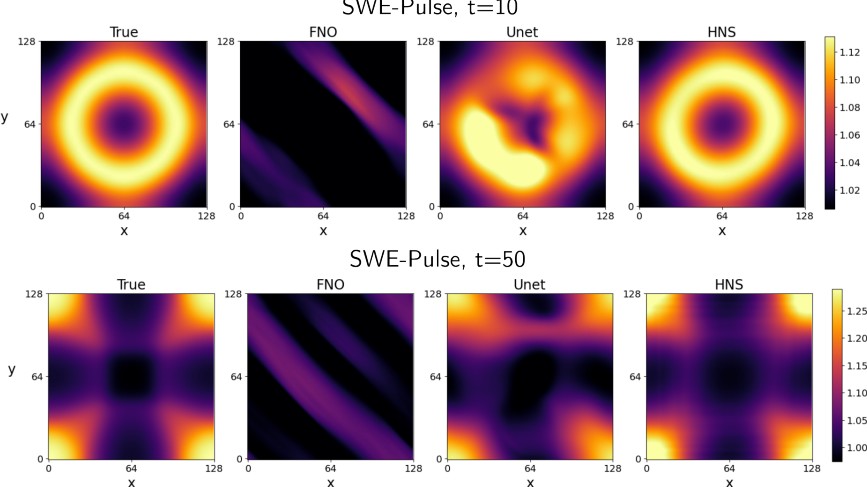

**Figure 7:** Predicted SWE trajectories with Gaussian pulse ICs, displayed at $t = 10$ and $t = 50$. Despite showing different behavior, HNS can better generalize to OOD samples.

# B Additional Results

## B.1 Hamiltonian Errors and Inductive Bias

| Model | Adv | KdV | SWE-Sines | SWE-Pulse |
|---|---|---|---|---|
| FNO | 2.18 | NaN | 0.0091 | 0.1326 |
| Unet | 0.22 | 2.37 | 0.0053 | 0.0158 |
| FNO($\frac{d\mathbf{u}}{dt}$) | 0.033 | 6.75e8 | - | - |
| Unet($\frac{d\mathbf{u}}{dt}$) | 0.043 | 5.76e6 | - | - |
| HNS | 0.0002 | 1.32 | 0.0015 | 0.0003 |

**Table 4:** Relative L2 Error for each experiment, evaluated on the Hamiltonian of predicted trajectories. Despite not including the Hamiltonian in the training loss and testing on OOD samples, HNS is exceptional at predicting solutions that conserve the Hamiltonian.

In the main body, we presented qualitative evidence that HNS is able to better conserve the Hamiltonian of predicted trajectories. To quantify this claim, we calculate the Relative L2 Error of the Hamiltonian of predicted trajectories across all PDE experiments. Specifically, the error of a sample is given by: $\frac{1}{T}\sum_{t=1}^{T}\frac{||\mathcal{H}[\mathbf{u}^t]-\mathcal{H}[\mathbf{u}_\theta^t]||_2^2}{||\mathcal{H}[\mathbf{u}^t]||_2^2}$, where $\mathbf{u}^t$ is the true solution at time $t$, $\mathbf{u}_\theta^t$ is the predicted solution at time $t$, and $\mathcal{H}[\cdot]$ is the Hamiltonian given by the integrals in Equations 8 and 12. The Hamiltonian errors are averaged across the validation sets and reported in Table 4. We note that in the KdV experiments, the dynamics are chaotic and the Hamiltonian is highly nonlinear, both of which contribute to large errors once predicted trajectories diverge. Overall, we find that HNS has exceptional capabilities in implicitly conserving energy-like quantities in their prediction.

Interestingly, this capability exists even when not trained with a loss on the Hamiltonian (i.e., $||\mathcal{H}[\mathbf{u}^t]-\mathcal{H}[\mathbf{u}_\theta^t]||_2^2$) and when predicting out-of-distribution trajectories. Where does this inductive bias to conserve the Hamiltonian come from? Based on Gruver et al. [42], we hypothesize four inductive biases in HNS that may contribute to this:

1. ODE Bias: HNS predicts $\frac{d\mathbf{u}}{dt}$ and uses an ODE integrator to evolve PDE dynamics.

2. Hamiltonian Bias: HNS relies on $\mathcal{J}\frac{\delta\mathcal{H}}{\delta\mathbf{u}}$ to calculate $\frac{d\mathbf{u}}{dt}$.

3. Gradient Learning Bias: HNS relies on autograd($\mathcal{H}_\theta[\mathbf{u}], \mathbf{u}$) to calculate $\frac{\delta\mathcal{H}}{\delta\mathbf{u}}$.

4. Neural Functional Bias: HNS relies on integral kernel functionals to calculate autograd($\mathcal{H}_\theta[\mathbf{u}], \mathbf{u}$).

We note that these biases are listed from simplest to most complex and are cumulative; for example, it is not possible to implement Hamiltonian structure without also implementing an ODE integrator. We modify a Unet to incorporate each inductive bias and evaluate its performance on the KdV equation, which is the most chaotic setting. The ODE bias was considered in the main work by training $d\mathbf{u}/dt$ model variants, while incorporating Hamiltonian bias is done by changing the training label to $\delta\mathcal{H}/\delta\mathbf{u}$ and using $\mathcal{J}$ during inference. To extend Unets to gradient learning, we add a linear head to project the output field to a scalar and use automatic differentiation to obtain $\delta\mathcal{H}/\delta\mathbf{u}$. After training, we evaluate model correlation time and Hamiltonian error on the validation set and the results are reported in Table 5.

We find that the main contributors to HNS performance and its ability to conserve energies over time is using Hamiltonian structure as well as the neural functional architecture. Using an ODE integrator alone degrades model performance and implementing gradient learning with Unets also harms performance. Interestingly, using gradient learning with conventional architectures disproportionately harms energy conservation; this implies that in PDEs, gradient domain learning improves energy conservation only when architectures have meaningful functional derivatives. This is not readily apparent, as observations from molecular dynamics or HNNs suggest that gradient domain learning generally improves energy conservation without considering the underlying architectures [35, 46, 48]. Based on these findings, we hypothesize the following mechanism underlying HNS: integral kernel functionals enable the learning of functional derivatives, combining this with infinite-dimensional Hamiltonian mechanics leads to its performance and conservation properties.

| Metric: | Correlation Time ($\uparrow$) | Hamiltonian Error ($\downarrow$) |
|---|---|---|
| Unet (Base) | 125.25 | 2.37 |
| Unet (ODE) | 120.5 | 5.76e6 |
| Unet (Ham.) | 143.75 | 2.06 |
| Unet (Grad.) | 141 | 4.71 |
| HNS | 151.75 | 1.32 |

**Table 5:** Correlation time and Hamiltonian errors for Unet models with increasingly more inductive biases, compared to HNS on the KdV equation. Using ODE integrators or gradient domain learning both degrade performance, while using Hamiltonian structure or neural functionals both increase performance and energy conservation.

## B.2 Timing Experiments

| Model (#Params) | Adv | KdV | SWE-Sines |
|---|---|---|---|
| FNO (65K/135K/7M) | 0.083 | 0.091 | 0.967 |
| Unet (65K/146K/3M) | 0.138 | 0.146 | 1.345 |
| HNS (32K/87K/3M) | 0.126 | 0.228 | 4.547 |
| Numerical Solver | 0.000 | 110.9 | 33.26 |

**Table 6:** Computational cost per inference step for each model across different PDEs, given in milliseconds (ms). Model sizes are also reported for the (Adv/Kdv/SWE) experiments. Each run uses a single NVIDIA RTX 6000 Ada GPU. Numerical solvers are run on a AMD Ryzen Threadripper PRO 5975WX 32-Core CPU due to lack of GPU compatibility.

The additional overhead of using automatic differentiation, evaluating the operator $\mathcal{J}$, and applying an ODE integrator during inference increases the computational cost of HNS. We examine this in Table 6 by performing a model rollout and averaging the time per inference step across a batch. Additionally, we report the runtimes of the numerical solvers used. In the case of 1D Advection, the analytical solution is used. Randomly sampled initial conditions can cause variability in the stiffness of the solution, therefore, solver runtimes are averaged across 10 samples. We note that the 2D SWE solver (PyClaw) [31] uses a Fortran compiler and is optimized for hyperbolic PDEs, while the KdV solver is purely Python-based. We note that for a fairer comparison, the numerical solvers should be coarsened to match the accuracy of the neural solvers.

Consistent with prior works, FNO models remain a fast neural PDE solver. While HNS may have a lower parameter count, this may be a misleading measure of inference speed. Automatic differentiation calculates the gradient with respect to each input point, therefore the computational cost of this step can scale with the size of the input grid. This can be seen when comparing the Adv and KdV experiments, where the grid size doubles from $n_x = 128$ to $n_x = 256$, and the computational cost roughly doubles as well. In 2D experiments, the cost of HNS is appreciably more than other baselines, although it is still modestly faster than an optimized numerical solver. We do not anticipate that the finite-difference (FD) schemes used in $\mathcal{J}$ or ODE integrators contribute meaningfully to computational cost, since FD schemes use $\mathcal{O}(M)$ operations, where $M$ is the number of grid points, and numerical integration occurs in constant time.

## B.3 Ablation Studies

There are many choices for parameterizing a kernel integral, such as from different kernels (linear, nonlinear), different architectures (MLP, SIREN), different conditioning mechanisms (concat/FiLM) and different receptive fields (local, global). Since this design space is large, we consider the effects of incrementally improving each aspect and report the performance of different HNS models in Table 7. We find that a nonlinear kernel is needed in all cases, since all the Hamiltonians tested are nonlinear. In addition, the use of FiLM conditioning and SIRENs tend to improve performance. Lastly, global conditioning is necessary for the KdV equation, which has nonlocal terms ($\frac{\partial^2 u}{\partial x^2}$) in the Hamiltonian.

| Model | Rollout Error | Model | Correlation Time | Model | Rollout Error |
|---|---|---|---|---|---|
| Base | 1.248 | Base | 51.75 | Base | 0.113 |
| + Nonlinear | 0.028 | + Nonlinear | 73.00 | + Nonlinear | 0.034 |
| + FiLM | 0.016 | + FiLM | 65.25 | + FiLM | 0.030 |
| + SIREN | 0.003 | + SIREN | 73.25 | + SIREN | 0.026 |
| + Global | 0.057 | + Global | 151.75 | + Global | 0.024 |
| **(a)** Ablations for 1D Advection | | **(b)** Ablations for 1D KdV | | **(c)** Ablations for 2D SWE-Sines | |

**Table 7:** Ablation studies for the HNS architecture. The base architecture is specified by (Linear, Concat, SIREN, Local). Each successive row is cumulative, adding an additional feature to the model. Parameter sizes are approximately constant across ablations.

## B.4 Varying Dataset Sizes

We study the effect of varying the number of samples in the training dataset on HNS performance in Table 8. The training dataset is truncated by a factor of 8, 4, or 2, and evaluated on the same validation dataset. This is repeated across all experiments/PDEs. In general, increasing the dataset size consistently improves model performance. This is more acute for 1D equations. For 2D SWE, the extra spatial dimension may allow for more data per sample relative to the complexity of the dynamics.

| Dataset Size | Adv | KdV | SWE-Sines | SWE-Pulse |
|---|---|---|---|---|
| $f$=1/8 | 1.320 | 71.00 | 0.0293 | 0.0243 |
| $f$=1/4 | 0.302 | 71.25 | 0.0274 | 0.0230 |
| $f$=1/2 | 0.075 | 95.25 | 0.0254 | 0.0227 |
| $f$=1 | 0.003 | 151.75 | 0.0249 | 0.0216 |

**Table 8:** Rollout error ($\downarrow$) or correlation times ($\uparrow$) of HNS models trained on datasets downsampled by a factor of $f$ (e.g., $f = 1/2$ uses half the number of training samples in the training set).

## B.5 Discretization Invariance

We experiment with querying models at different discretizations than its training resolution. The results are in Table 9, with the training resolution in bold. Rollout errors are averaged over the validation set, where validation labels are either downsampled or re-solved at a higher resolution using the same initial conditions. Additionally, for 1D Advection, we conducted an experiment in which the model is queried on an unseen grid $x = [0, 32], n_x = 256$, after being trained on a grid $x = [0, 16], n_x = 128$, denoted as $256^*$.

**(a)** Rollout Error at Different Resolutions, 1D Adv

**(b)** Rollout Error at Different Resolutions, 2D SWE (Sines)

**(c)** Rollout Error at Different Resolutions, 2D SWE (Pulse)

| Resolution | HNF | FNO | Unet | Resolution | HNF | FNO | Unet | Resolution | HNF | FNO | Unet |
|---|---|---|---|---|---|---|---|---|---|---|---|
| 64 | 0.0029 | 0.064 | 1.19 | (64, 64) | 0.028 | 0.066 | 0.098 | (64, 64) | 0.035 | 0.173 | 0.173 |
| **128** | 0.0033 | 0.064 | 0.075 | **(128, 128)** | 0.029 | 0.066 | 0.008 | **(128, 128)** | 0.024 | 0.174 | 0.117 |
| 256 | 0.0055 | 0.050 | 1.32 | (256, 256) | 0.026 | 0.066 | 0.110 | (256, 256) | 0.019 | 0.174 | 0.225 |
| 256* | 0.0021 | 1.34 | 0.072 | | | | | | | | |

**Table 9:** Rollout error of models at different resolutions for 1D Adv and 2D SWE datasets.

In all cases, HNFs are capable of zero-shot super-resolution, as is possible with Neural Operators. Additionally, the error is approximately constant across discretizations and on a new, unseen grid for 1D Advection. Predictably, FNO models have nearly constant error across discretizations. Interestingly, for an extrapolated grid in 1D Advection, FNO struggles to make predictions. For Unet

models, the error increases when the grid spacing changes, however for a constant spacing, it is able to extrapolate to unseen grids (i.e. $[0, 16] \to [0, 32]$ with $dx$ unchanged).

## C Additional Theory

### C.1 Proof of Linear Functional Approximation

Interestingly, while development of neural functionals was arrived upon independently, a proof of linear functional approximation exists. In fact, approximating a linear functional with an integral kernel is an intermediate step in operator approximation theorems [77–79] and its proof comes as a series of lemmas in Kovachki et al. [9]. For the sake of completeness, we provide relevant definitions and cite the necessary lemmas here.

**Definitions (Appendix A, Kovachki et al. [9])** Let $D \subset \mathbb{R}^d$ be a domain. Let $\mathbb{N}_0 = \mathbb{N} \cup \{0\}$ denote the natural numbers that include zero. Let $\mathcal{X}$ be a Banach space, and $\mathcal{X}^*$ be its continuous dual space. Specifically, the dual space contains all continuous, linear functionals $f : \mathcal{X} \to \mathbb{R}$ with the norm: $||f||_{\mathcal{X}^*} = \sup_{x \in \mathcal{X}, ||x||_{\mathcal{X}} = 1} |f(x)| < \infty$.

For any multi-index $\alpha \in \mathbb{N}_0^d$, $\partial^\alpha$ is the $\alpha$-th partial derivative of $f$. The following spaces are defined for $m \in \mathbb{N}_0$:

- $C(D) = \{f : D \to \mathbb{R} : f \text{ is continuous}\}$
- $C^m(D) = \{f : D \to \mathbb{R} : \partial^\alpha f \in C^{m-|\alpha|_1}(D) \, \forall \, 0 \le |\alpha|_1 \le m\}$
- $C_c^\infty(D) = \{f \in C^\infty(D) : \text{supp}(f) \subset D \text{ is compact}\}$
- $C_b^m(D) = \{f \in C^m(D) : \max_{0 \le |\alpha|_1 \le m} \sup_{x \in D} |\partial^\alpha f(x)| < \infty\}$
- $C^m(\bar{D}) = \{f \in C_b^m(D) : \partial^\alpha f \text{ is uniformly continuous } \forall 0 \le |\alpha|_1 \le m\}$

Additionally, we cite definitions of neural networks from Section 8.1, Kovachki et al. [9]. For any $n \in \mathbb{N}$ and $\sigma : \mathbb{R} \to \mathbb{R}$, a neural network with $n$ layers is given by:

$$
\begin{aligned}
N_n(\sigma; \mathbb{R}^d, \mathbb{R}^{d'}) := \{f : \mathbb{R}^d \to \mathbb{R}^{d'} : & f(x) = W_n \sigma(\dots W_1 \sigma(W_0 x + b_0) + b_1 \dots) + b_n, \\
& W_0 \in \mathbb{R}^{d_0 \times d}, W_1 \in \mathbb{R}^{d_1 \times d_0}, \dots, W_n \in \mathbb{R}^{d' \times d_{n-1}}, \\
& b_0 \in \mathbb{R}^{d_0}, b_1 \in \mathbb{R}^{d_1}, \dots, b_n \in \mathbb{R}^{d'}, d_0, d_1, \dots, d' \in \mathbb{N}\}
\end{aligned}
$$

The activation functions $\sigma$ are restricted to the set:

$$
\sigma \in A_m := \{\sigma \in C(\mathbb{R}) : \exists n \in \mathbb{N} \text{ s.t. } N_n \text{ is dense in } C^m(K) \forall K \subset \mathbb{R}^d \text{ compact}\}
$$

to allow universal approximation. Lastly, functions are restricted to be real-valued, although extensions to vector-valued functions are possible (Section 8.3, Kovachki et al. [9]).

**Theorem C.1** (Lemma 30, Kovachki et al. [9]). *Let $D \subset \mathbb{R}^d$ be a domain and $L \in (C(\bar{D}))^*$ be a linear functional. For any compact set $K \subset C(\bar{D})$ and $\epsilon > 0$, there exists a function $\kappa \in C_c^\infty(D)$ such that:*

$$
\sup_{u \in K} |L(u) - \int_D \kappa u \, dx| < \epsilon
$$

This result establishes that a linear functional acting on functions in $C(\bar{D})$ can be approximated by a smooth integral kernel with arbitrary error $\epsilon$. To show functional approximation with a neural network, we slightly modify an existing lemma:

**Theorem C.2** (Lemma 32, Kovachki et al. [9]). *Let $D \subset \mathbb{R}^d$ be a domain and $\mathcal{A} = C(\bar{D})$. Let $L \in \mathcal{A}^*$ be a linear functional. For any compact set $K \subset \mathcal{A}$, $\sigma \in A_0$, and $\epsilon > 0$, there exists a number $L \in \mathbb{N}$ and neural network $\kappa_\theta \in N_L(\sigma; \mathbb{R}^d, \mathbb{R})$ such that:*

$$
\sup_{u \in K} |L(u) - \int_D \kappa_\theta(x) u(x) \, dx|_1 < \epsilon
$$

*Proof.* Since $K$ is bounded, there exists a number $M > 0$ such that:

$$\sup_{u \in K} ||u||_{\mathcal{A}} \leq M$$

By Lemma 30, there exists a function $\kappa \in C_c^\infty(D)$ such that:

$$\sup_{u \in K} |L(u) - \int_D \kappa u \mathrm{d}x| < \frac{\epsilon}{2}$$

Since $\sigma \in A_0$, there exists some $L \in \mathbb{N}$ and a neural network $\kappa_\theta \in N_L(\sigma; \mathbb{R}^d, \mathbb{R})$ such that:

$$||\kappa_\theta - \kappa||_C \leq \frac{\epsilon}{2M|D|}$$

Then for any $u \in K$:

$$
\begin{aligned}
|L(u) - \int_D \kappa_\theta(x) u(x) \mathrm{d}x|_1 &\leq |L(u) - \int_D \kappa u \mathrm{d}x|_1 + |\int_D (\kappa - \kappa_\theta) u \mathrm{d}x|_1 \\
&\leq \frac{\epsilon}{2} + M|D| * ||\kappa - \kappa_\theta||_C \\
&\leq \epsilon
\end{aligned}
$$

$\square$

Therefore, linear functionals for continuously differentiable function classes can be approximated to arbitrary accuracy by a neural network using an integral kernel. Extensions to Sobolev and Hilbert spaces can also be proved in a similar manner.

Interestingly, this overlap between neural functionals and neural operators suggests a deeper connection between the two. The integral kernel functional bears many similarities to the integral kernel operator and, in practice, is implemented as an integral kernel operator evaluated at a single output. From an implementation perspective, the two architectures are equivalent in this manner; examining the theoretical basis also suggests that neural functionals are a subset of neural operators when approximating linear functionals. This can be seen by looking at operator approximation theorems (Lemma 22, Kovachki et al. [9]). We refrain from rewriting the theorems here, but focus on the necessary modifications. Specifically, to use a neural operator to approximate a linear functional, the output Banach space $\mathcal{Y}$ is set to $\mathbb{R}$ and intermediate dimensions $J, J'$ are set to 1. The linear map $F_J : \mathcal{X} \to \mathbb{R}^J$ from the input field $x \in \mathcal{X}$ becomes $F_J = w_1(x)$, where $w_1(x) \in \mathcal{X}^*$. Lastly, the finite-dimensional map is defined as $\varphi \in C(\mathbb{R}, \mathbb{R})$ and the output map is defined as $G_{J'} : \mathbb{R} \to \mathbb{R}$.

### C.2 Connection to the Cylindrical Approximation

Miyagawa and Yokota [56] propose an alternate mechanism to approximate functionals based on the cylindrical approximation. We explore a possible connection with the current method of using an integral kernel.

We consider functionals $F[u] : H \to \mathbb{R}$, where we restrict functions $u(x) \in H$ to be in a Hilbert space $H$ with inner product $\langle \cdot, \cdot \rangle_H$. Since functions can be represented as a sum of basis functions, $u(x)$ can be written as $u(x) = \sum_{k=1}^\infty \langle u, \phi_k \rangle \phi_k(x)$, with an orthonormal basis $\{\phi_1, \phi_2, \ldots\}$. Truncating the sum to a finite number of terms $m$ allows the cylindrical approximation of functionals [80–82]. This is obtained by substituting the truncated series into the functional:

$$f(a_1, a_2, \ldots, a_m) := F[\sum_{k=1}^m a_k \phi_k(x))] \tag{15}$$

where $a_k$ are basis coefficients $\langle u, \phi_k \rangle$. Using the cylindrical approximation $f$, functional derivatives can be approximated by:

$$\frac{\delta F}{\delta u} \approx \sum_{k=1}^m \frac{\partial f}{\partial a_k} \phi_k(x) \tag{16}$$

which converges to the true functional derivative as $m \to \infty$ [81]. When discretizing the functional derivative, we define a set of $n$ points $\mathbf{x} = [x_1, x_2, \ldots, x_n]$ at which the basis functions are evaluated, which results in the discrete approximation of the functional derivative:

$$\sum_{k=1}^{m} \frac{\partial f}{\partial a_k} \phi_k(x) \approx \sum_{k=1}^{m} \frac{\partial f}{\partial a_k} \phi_k(\mathbf{x}) = \begin{bmatrix} \sum \frac{\partial f}{\partial a_k} \phi_k(x_1) \\ \sum \frac{\partial f}{\partial a_k} \phi_k(x_2) \\ \vdots \\ \sum \frac{\partial f}{\partial a_k} \phi_k(x_n) \end{bmatrix} \tag{17}$$

We are interested if the gradient of a neural functional that is parameterized by a kernel integral can represent this discrete object, and what conditions are needed to do so. Furthermore, there are two levels of approximation (a truncated basis and a discretized domain) and we are interested if the gradient of a neural functional can recover a functional derivative in its limit. We will work through three cases of increasing complexity, using a linear kernel, a nonlinear kernel, and a nonlinear and global kernel.

**Linear Kernel Integrals**    A functional parameterized by a linear kernel integral can be written as:

$$F_\theta[u] = \int_\Omega \kappa_\theta(x) u(x) \mathrm{d}x \approx \sum_{i=1}^{n} \kappa_\theta(x_i) u_i \mu_i \tag{18}$$

using a Riemann sum approximation and quadrature weights $\mu_i$. Without loss of generality, we can assume some constant quadrature weight $\mu$ and write the gradient of the Riemann sum as:

$$\mu \nabla_u \left( \sum_{i=1}^{n} \kappa_\theta(x_i) u_i \right) = \mu \begin{bmatrix} \frac{\partial}{\partial u_1}(\kappa_\theta(x_1) u_1) \\ \frac{\partial}{\partial u_2}(\kappa_\theta(x_2) u_2) \\ \vdots \\ \frac{\partial}{\partial u_n}(\kappa_\theta(x_n) u_n)) \end{bmatrix} = \mu \begin{bmatrix} \kappa_\theta(x_1) \\ \kappa_\theta(x_2) \\ \vdots \\ \kappa_\theta(x_n) \end{bmatrix} \tag{19}$$

We can interpret this gradient as the discretization of a single basis function $\kappa_\theta = \phi_1$, and without the necessary coefficients $\frac{\partial f}{\partial a_k}$. Evidently, this is not enough expressivity to approach a sum of $m$ coefficients and basis functions, even as $n \to \infty$. Intuitively, the output $F_\theta[u]$ does not depend on $u$ enough, causing its gradient $\nabla_u$ to be too simple. A potential remedy is to modify the kernel to be nonlinear by using $\kappa_\theta(x, u(x))$.

**Nonlinear Kernel Integrals**    Following the same approximation, the gradient of a nonlinear kernel integral can be written as:

$$\mu \nabla_u \left( \sum_{i=1}^{n} \kappa_\theta(x_i, u_i) u_i \right) = \mu \begin{bmatrix} \frac{\partial}{\partial u_1}(\kappa_\theta(x_1, u_1) u_1) \\ \frac{\partial}{\partial u_2}(\kappa_\theta(x_2, u_2) u_2) \\ \vdots \\ \frac{\partial}{\partial u_n}(\kappa_\theta(x_n, u_n) u_n)) \end{bmatrix} = \mu \begin{bmatrix} u_1 \frac{\partial}{\partial u_1} \kappa_\theta(x_1, u_1) + \kappa_\theta(x_1, u_1) \\ u_2 \frac{\partial}{\partial u_2} \kappa_\theta(x_2, u_2) + \kappa_\theta(x_2, u_2) \\ \vdots \\ u_n \frac{\partial}{\partial u_n} \kappa_\theta(x_n, u_n) + \kappa_\theta(x_n, u_n) \end{bmatrix} \tag{20}$$

using the product rule. In general, $\kappa_\theta(x_i, u_i)$ can be quite complex and nonlinear, however, by construction it only depends on $(x_i, u_i)$. We define a function $d_i(x_i) = \frac{\partial}{\partial u_i} \kappa_\theta(x_i, u_i)|_{u_i=u_i}$, which evaluates the partial derivative at $x_i$. This lets us construct at most $n$ basis functions $\{d_1, d_2, \ldots, d_n\}$, however, this is still not enough expressivity, as each entry of the discretized functional derivative is still restricted to one basis function, albeit a different one at each row. Despite this shortcoming, writing this expansion allows us to see that producing a sum over multiple basis functions requires the kernel to depend globally on $\mathbf{u} = [u_1, u_2, \ldots, u_n]$, since the gradient of the Riemann sum no longer degenerates to a single term at each row.

**Nonlinear Global Kernel Integrals**   Using the same approximation, the gradient of a nonlinear, global kernel integral can be written as:

$$\mu \nabla_u (\sum_{i=1}^{n} \kappa_\theta(x_i, \mathbf{u}) u_i) = \mu \begin{bmatrix} \frac{\partial}{\partial u_1}(\sum \kappa_\theta(x_i, \mathbf{u}) u_i) \\ \frac{\partial}{\partial u_2}(\sum \kappa_\theta(x_i, \mathbf{u}) u_i) \\ \vdots \\ \frac{\partial}{\partial u_n}(\sum \kappa_\theta(x_i, \mathbf{u}) u_i) \end{bmatrix} \tag{21}$$

The expression $\frac{\partial}{\partial u_j}(\sum \kappa_\theta(x_i, \mathbf{u}) u_i)$ can be expanded into:

$$\frac{\partial}{\partial u_j}(\sum_{i=1}^{n} \kappa_\theta(x_i, \mathbf{u}) u_i) = u_1 \frac{\partial}{\partial u_j} \kappa_\theta(x_1, \mathbf{u}) + \dots \tag{22}$$

$$+ u_j \frac{\partial}{\partial u_j} \kappa_\theta(x_j, \mathbf{u}) + \kappa_\theta(x_j, \mathbf{u}) + \dots$$

$$+ u_n \frac{\partial}{\partial u_j} \kappa_\theta(x_n, \mathbf{u})$$

In general, we can construct a kernel $\kappa_\theta$ such that $\frac{\partial}{\partial u_j} \kappa_\theta(x_i, \mathbf{u})$ is different for $i = \{1, \dots, n\}$. A simple example is provided based on FiLM modulation:

$$\kappa_\theta(x_i, \mathbf{u}) = [W^u \mathbf{u}]_i (W^x x_i + b^x) + [b^u]_i \tag{23}$$

where $W^u \in \mathbb{R}^{n \times n}, W^x \in \mathbb{R}^{1x1}, b^u \in \mathbb{R}^n, b^x \in \mathbb{R}^1$ and $[\cdot]_i$ is an operation that takes the $i$-th row of a column vector. In this case, $\frac{\partial}{\partial u_j} \kappa_\theta(x_i, \mathbf{u})$ is equivalent to a function $d_{ij}(x_i) = W^q_{ij} W^x x_i$. With even deeper and more complex kernels, functions $d_{ij}$ can become more expressive. Without loss of generality, we can also absorb the basis coefficients into $d_{ij}$.

With this formulation, we can see that the $j$-th row of the gradient vector can be written as a sum of $n$ functions $\sum_{i=1}^{n} d_{ij}(x_i)$, which reduces to the cylindrical approximation when letting $d_{ij}$ be arbitrary. Furthermore, letting $n \to \infty$ increases the number of functions $d_{ij}$ and coordinates $x_i$, and is consistent with the interpretation of simultaneously increasing the number of bases to approach the true functional derivative as well as converting from a discrete to continuous representation.

The cylindrical approximation makes no assumption on the linearity of $F$, therefore, when using a global, nonlinear kernel, a kernel integral can represent a functional derivative. This theoretical insight is also supported by empirical observations. Through ablation studies in Section B.3, we find that modeling complex functional derivatives (such as in the KdV equation) is only possible with a global, nonlinear kernel, and simplifications of the kernel degrade the performance.

## C.3   Derivations with Varying Coefficients and Source Terms

Using HNS with PDEs with varying coefficients and source terms is possible on a case-by-case basis, requiring additional effort to derive the Hamiltonian structure. We provide examples below by introducing coefficients to 1D Advection and 1D KdV as well as varying bathymetry (source terms) to 2D SWE.

**Varying Coefficients**   In the presence of coefficients, the Hamiltonians for 1D Advection and 1D KdV become:

$$\mathcal{H}_{adv}[u] = \int_\Omega -\frac{c}{2}(u(x))^2 \mathrm{d}x, \qquad \mathcal{H}_{kdv}[u] = \int_\Omega -\frac{\alpha}{6}(u(x))^3 - \beta u(x) \frac{\partial^2 u}{\partial x^2}(x) \mathrm{d}x, \tag{24}$$

For 1D Advection, one can verify the Hamiltonian structure:

$$\frac{\delta \mathcal{H}_{adv}}{\delta u} = -c u(x), \qquad \mathcal{J}(\frac{\delta \mathcal{H}_{adv}}{\delta u}) = -c \frac{\partial u}{\partial x} = \frac{\mathrm{d}u}{\mathrm{d}t} \tag{25}$$

A similar calculation recovers the PDE for the 1D KdV equation ($u_t + \alpha u u_x + \beta u_{xxx} = 0$) from Hamilton's equations by using $\frac{\delta \mathcal{H}_{kdv}}{\delta u} = -\frac{\alpha}{2} u^2 - \beta u_{xx}$. To accommodate varying coefficients, this suggests training HNS with coefficient information using the loss: $\mathcal{L}(\nabla_u \mathcal{H}_\theta(\mathbf{u}, \mathbf{x}, \mathbf{c}), \frac{\delta \mathcal{H}}{\delta u})$. In special cases where the coefficients can be factored out (i.e., $\frac{\delta \mathcal{H}[u,c]}{\delta u} = c \frac{\delta \mathcal{H}[u]}{\delta u}$), once can scale the base model output $\nabla_u \mathcal{H}_\theta(\mathbf{u}, \mathbf{x})$ by $c$ to achieve the same effect.

**Varying Source Terms**  The 2D shallow-water equations describe free surfaces where the horizontal length scale is significantly larger than the vertical length scale. Within this setup, the free surface can be above a spatially-varying bathymetry $b(x, y)$, or the topography of the bottom surface. Changes in the elevation of the bathymetry can accelerate or decelerate flows, which acts as a source term on the momentum balance [83, 84]. In particular, the shallow-water equations are modified to become:

$$\partial_t h + \nabla \cdot (\mathbf{v}h) = 0, \qquad \partial_t \mathbf{v} + \mathbf{v} \cdot \nabla \mathbf{v} = -g\nabla h - gh\nabla b \tag{26}$$

Furthermore, $h(x, y, t)$ now represents the height above the bathymetry $b(x, y, t)$. This source term is represented in the Hamiltonian as:

$$\mathcal{H}^b[\mathbf{u}] = \mathcal{H}^b[v_x, v_y, h] = \int_\Omega \frac{1}{2}h(v_x^2 + v_y^2) + \frac{1}{2}gh^2(1 + b) \, \mathrm{d}A \tag{27}$$

and the operator matrix $\mathcal{J}$ is left unchanged (Equation 11). One can check that applying Hamilton's equations recovers the modified shallow-water equations. When comparing this derivation to one with constant bathymetry, one can show that:

$$\frac{\delta \mathcal{H}^b[u]}{\delta u} = \frac{\delta \mathcal{H}[u]}{\delta u} + \begin{bmatrix} 0 \\ 0 \\ ghb \end{bmatrix} \tag{28}$$

This suggests that instead of scaling model outputs $\nabla_u \mathcal{H}_\theta(\mathbf{u}, \mathbf{x})$ (as in the case of coefficients), source terms can be potentially captured by adding terms to model outputs. In PDEs where a decomposition like this is not possible, the source term will need to be added to the model input to produce the output $\nabla_u \mathcal{H}_\theta(\mathbf{u}, \mathbf{x}, \mathbf{b})$, where bold terms indicate discretized fields/coordinates.

# D    Implementation Details

## D.1    Numerical Methods

**Numerical Integration**  ODE integration is performed using a 2nd-order Adams-Bashforth scheme. The solution at the next timestep $\mathbf{u}^{t+1}$ is calculated using the current timestep $\mathbf{u}^t$ and an estimate of the current derivative $\frac{d\mathbf{u}}{dt}|_{t=t} = f(\mathbf{u}^t)$:

$$\mathbf{u}^{t+1} = \mathbf{u}^t + \Delta t \left( \frac{3}{2}f(\mathbf{u}^t) - \frac{1}{2}f(\mathbf{u}^{t-1}) \right) \tag{29}$$

This can be implemented in constant time by caching prior derivative estimates $\{f(\mathbf{u}^i) : i < t\}$, allowing the ODE integrator to have a 2nd-order truncation error ($\mathcal{O}(\Delta t^2)$) while remaining fast. At the first timestep, there are no cached derivatives, therefore a first-order Euler integrator is used: $\mathbf{u}^{t+1} = \mathbf{u}^t + \Delta t f(\mathbf{u}^t)$. For the given experiments, these integrators are accurate since $\Delta t$ is small enough; to be used with larger timesteps $\Delta t$, higher-order integrators may be needed or smaller, intermediate steps will need to be taken [26]. This adds additional consideration to the training and deployment of derivative-based neural PDE solvers.

**Finite Difference Schemes**  Implementing infinite-dimensional Hamiltonian structure involves approximating the operator $\mathcal{J}$, either through numerical methods or neural operators. We opt for numerical methods, as the operators are simple and in all our tests only involve first-order spatial derivatives $\partial_x$. For the 1D Advection equation, we implement a 2nd-order central difference scheme. For the 1D KdV equation, the spatial derivatives exhibit larger gradients, therefore we use an 8th-order central difference scheme with a smoothing stencil to approximate $\partial_x$ [85]. A similar scheme is implemented in 2D to approximate $\partial_x$ and $\partial_y$ for the 2D SWE experiments, and additionally, a Savitzky-Golay filter is used to further damp numerical oscillations. These arise since the lack of viscosity creates discontinuities in the shallow-water equations, alternative methods may include flux-limiting or non-oscillatory schemes [86, 87]. While these implementations work, they still required tuning, as such, the use of a neural operator to approximate $\mathcal{J}$ to make HNS fully learnable would be an interesting future direction.

Additionally, to calculate temporal derivatives $d\mathbf{u}/dt$ during training, a 4-th order Richardson extrapolation is used to provide accurate derivatives. At the boundaries $t = 0$, $t = T$ a one-sided Richardson extrapolation is used [88].

**Numerical Solvers**    The analytical solution to the Advection equation is used to generate its datasets. The numerical solver for the KdV equation is from Brandstetter et al. [24]; due to the stiffness of the PDE, the solutions do not always conserve the Hamiltonian. The WENO schemes used allow for more stable derivatives, but at the cost of numerical viscosity, which can affect the Hamiltonian since it is highly nonlinear. Therefore, we filter the generated KdV solutions and extract 50% of the trajectories with the smallest variation in the Hamiltonian. Similar fluctuations in the Hamiltonian are seen in SWE trajectories generated from PyClaw [31], but the variations are small enough to be ignored.

## D.2    Experimental Details

**Hyperparameters**    Each model has different hyperparameters for different PDEs (Adv/KdV/SWE). We provide key hyperparameters for these experiments in Tables 10, 11, 12.

|        | Adv | KdV | SWE |
|--------|-----|-----|-----|
| Modes  | 10  | 12  | 12  |
| Width  | 24  | 32  | 48  |
| Layers | 5   | 5   | 5   |

**Table 10:** FNO Parameters

|                | Adv | KdV | SWE |
|----------------|-----|-----|-----|
| Width          | 8   | 12  | 32  |
| Bottleneck Dim | 32  | 48  | 128 |
| Layers         | 6   | 6   | 6   |

**Table 11:** Unet Parameters

|        | Adv   | KdV    | SWE    |
|--------|-------|--------|--------|
| Width  | 32    | 32     | 64     |
| Kernel | local | global | global |
| Cond.  | FiLM  | FiLM   | FiLM   |

**Table 12:** HNS Parameters

**Computational Resources**    All experiments were run on a single NVIDIA RTX 6000 Ada GPU; 1D experiments were usually run within 10 minutes, while 2D experiments were usually run within an hour. The largest dataset is around 20 GB, for the 2D Shallow-Water equations.

