# OpenReview forum: "Hamiltonian Neural PDE Solvers through Functional Approximation"
_NeurIPS.cc/2025/Conference — NeurIPS 2025 poster_

### Official Review · Reviewer_a4TR · 2025-06-12

**Clarity:** 4
**Significance:** 3
**Originality:** 3
**Rating:** 4
**Confidence:** 4

**Summary:**

Building upon the Riesz theorem (Theorem 3.1), integral kernel operator (5), and neural field (7), the authors develop an architecture that represents linear (and potentially non-linear) functionals. A training algorithm to adapt it to functional Hamiltonian problems is also proposed. The experimental results demonstrate that the proposed model outperforms baseline models on several PDE tasks, while it is a bit computationally demanding.

**Questions:**

- [Question] Could you elaborate on the following statement and provide an example?

    > ... functional approximation and learning with its gradients may find other uses, such as in molecular dynamics or design optimization.  (lines 11-12)
    >
    > Beyond predicting (future) PDEs (states), functionals arise in many physical systems, opening interesting opportunities to apply and improve neural functionals for other meaningful uses. (lineas 41-42)

- [Question] (lines 118-119) Why did the authors adopt FiLM?

- [Question] Could the proposed model be extended to handle functional differential equations that involve functional derivatives?

- [Question] How robust are the experimental results to the grid size? Do you have any recommendations or practical insights on this?

**Ethical Concerns:**

["NO or VERY MINOR ethics concerns only"]

**Final Justification:**

The paper is well-written, and the main idea is clearly presented. The applicability of the proposed method to other domains is noteworthy. While the authors acknowledge the limitations of their theory for nonlinear functionals, addressing this would likely require a separate study, given the substantial challenges inherent in nonlinear functional analysis. Although the computational cost of the proposed model is higher than that of the baselines, it appears acceptable.

In the original submission, all reviewers raised concerns that several important prior studies were not cited.
Through the rebuttal, the authors clarified their contributions in relation to prior studies: Although this paper has an overlap in its scope with several prior studies, the authors develop their method on the Hamiltonian formulation, which can be an independent contribution of this paper.
Furthermore, the authors added error bars and increased the number of runs in their additional experimental results. Although some results do not appear statistically significant, the variances of the proposed method are sometimes smaller than baselines, which can be beneficial for practitioners. Additionally, the results help clarify the conditions under which the proposed method works well or not.
I believe that the paper's quality has significantly improved through the discussions with the reviewers, and the paper's contributions have been much clarified.

On the other hand, I am still concerned about the statistical significance of the experimental results because several error bars (signal-to-noise ratios) are large, and some of the results, if not all, are inconclusive, potentially affecting reproducibility of the work. Also, while the authors effectively addressed many concerns raised by the reviewers, the paper requires a major revision due to the aforementioned lack of citations, as outlined in the responses for all reviewers.

Overall, I think the contributions of the paper slightly outweigh the concerns, and I changed my score from 3 to 4 (Borderline accept).

**Limitations:**

Discussed in Section 5.

**Paper Formatting Concerns:**

No.

**Quality:**

3

**Strengths And Weaknesses:**

## Strengths

- (Technical Contribution) The combination of the integral kernel operator and neural field effectively solves functional Hamiltonian problems. Leveraging neural networks to simplify functional analysis is a promising direction, especially given recent advances in the numerical analysis of functionals.

- (Broader Impact) The underlying idea of the proposed model is simple and general and has the potential to be applied to a broader range of research areas, such as density functional theory (DFT) and functional differential equations.

- (Clarity) The paper is easy to follow and well-written. Background of the Hamiltonian mechanics, functional analysis, and neural field is provided, making the paper reader-friendly.

- (Reproducibility) The code is provided for reproducibility.


## Weaknesses

I hope my comments below help improve the quality of the work.

- The paper would benefit from a discussion of several related studies on neural functionals that are currently missing, which will highlight the authors’ contribution. To be clear, I am not suggesting the addition of extensive experimental comparisons with other baselines; rather, I would appreciate a discussion of the similarities and differences between the proposed method and prior work.

    - [Miyagawa & Yokota, 2024] Miyagawa and Yokota propose a neural architecture that can handle functional derivatives and solve functional PDEs. They also prove the convergence of their approximated solution converges to the true solution. They meticulously cite related studies and provide thorough discussions, which I think are helpful to highlight the contribution of the authors’ manuscript. I encourage the authors to compare their work with this study and discuss pros and cons.

    - [Zhou+, 2023a] [Zhou+, 2023b] Zhou et al. propose a neural functional that takes the weight parameters of DNNs as inputs and outputs vectors. This construction is particularly relevant to applications involving implicit neural representations (INRs). I encourage the authors to discuss the strengths and limitations of this neural functional in the context of their own work.

    - [Pederson+, 2022] [FIedler+, 2022] [Li+, 2023] Density functional theory (DFT) is an important application of functional theory. [Pederson+, 2022] and [FIedler+, 2022]  are review papers on DFT and machine learning. [Li+, 2023] is an example that apply deep learning to DFT. I encourage the authors to discuss a potential impact of the work on this field.

    - Please see also the references therein and add relevant papers to the manuscript.

- The mechanism by which functional derivatives can be efficiently computed via automatic differentiation appears to be a well-established approach; fundamentally, it stems from the discretization of spatial coordinates, where functional derivatives reduce to partial derivatives. This idea has long been employed in DFT, where functional derivatives are computed by discretizing spatial coordinates—reducing them to partial derivatives—and approximating them using neural networks (e.g., see [Meyer et al., 2020], [Costa et al., 2022], [Medvidović et al., 2025], and Appendix A in [Miyagawa & Yokota, 2024]). To my understanding, such approximations converge to the true functional derivative as the grid size decreases.

    - That said, the proposed method is specifically designed for solving PDEs, which is a perspective not typically addressed in DFT. I would recommend that the authors acknowledge this similarity and include a discussion. Relevant aspects for comparison include computational efficiency, convergence rate with respect to grid size, and solution accuracy.

- I would not recommend using the name “Neural Functional“ for the proposed architecture, as it has already used in several studies [Zhou+, 2023] and may be confusing (e.g., (liens 2-4) “In this work, we derive and propose a new architecture, the Neural Functional, which learns function to scalar mappings.” ([pdf](zotero://open-pdf/library/items/TMCVY4J3?page=1)) ).

- The following statements could come across as an overstatement:

    > (1) … no current neural architecture can theoretically model this infinite-dimensional to scalar mapping (=functional), let alone have meaningful functional derivatives. In this paper, we introduce the Neural Functional, an architecture that can provably represent functionals in the linear case… (lines 34-37)
    >
    > (2) … this is the first time an architecture has been developed to learn function to scalar (function-to-scalar) mappings. (lines 318-319)

    because (1) a neural architecture that can theoretically model functionals has proposed by [Miyagawa & Yokota, 2024], and (2) the neural operator can be used to map a function to a scalar. Strictly speaking, the model proposed by  [Miyagawa & Yokota, 2024] requires the number of basis functions $m$ and the neural network size $d$ to be large for accurate infinite-dimensional approximation; however, the authors’ neural functional also requires limit operation ($\Delta x\rightarrow0$ and $d \rightarrow \infty$ ) for provably accurate estimation of functionals (lines 102-158 & figure 1: approximating the integral and parameterizing the kernel).

    - (By the way, $m \rightarrow \infty$ in [Miyagawa & Yokota, 2024] corresponds to increasing spatial resolution and, at a high level, aligns with $\Delta x\rightarrow0$ in the authors’ framework, to my understanding.)

    - Again, the paper benefits from discussing the differences from [Miyagawa & Yokota, 2024].

- (Section 6: Related Works) I encourage the authors to clarify the advantage of the proposed model over other neural PDE surrogates [50-67].

- Error bars are missing in Tables 1, 4, 5, and 6.

- Error bars (standard deviations) in Tables 2 & 3 are not very informative, as they are based on only three trials. I would like to suggest increasing the number of runs, report all results, and/or use the box plot to provide a more comprehensive view of the variability..

- [Minor typo] (line 119) a -> an

- [Minor typo] (line 144) IN -> In

- [Minor comment] I recommend that the authors refrain from overusing semicolons, as it makes some sentences ambiguous.


## References

- [Miyagawa & Yokota, 2024] Miyagawa, Taiki, and Takeru Yokota. "Physics-informed Neural Networks for Functional Differential Equations: Cylindrical Approximation and Its Convergence Guarantees." Advances in Neural Information Processing Systems 37 (2024): 72274-72409.

- [Zhou+, 2023a] Zhou, Allan, et al. "Permutation equivariant neural functionals." Advances in neural information processing systems 36 (2023): 24966-24992.

- [Zhou+, 2023b] Zhou, Allan, et al. "Neural functional transformers." Advances in neural information processing systems 36 (2023): 77485-77502.

- [Pederson+, 2022] Pederson, Ryan, Bhupalee Kalita, and Kieron Burke. "Machine learning and density functional theory." Nature Reviews Physics 4.6 (2022): 357-358.

- [Fiedler+, 2022] Fiedler, Lenz, et al. "Deep dive into machine learning density functional theory for materials science and chemistry." Physical Review Materials 6.4 (2022): 040301.

- [Li+, 2023] Li, Tianbo, et al. "D4FT: A deep learning approach to Kohn-Sham density functional theory." ICLR 2023 (Top25%).

- [Meyer+, 2020] Meyer, Ralf, Manuel Weichselbaum, and Andreas W. Hauser. "Machine learning approaches toward orbital-free density functional theory: Simultaneous training on the kinetic energy density functional and its functional derivative." Journal of chemical theory and computation 16.9 (2020): 5685-5694.

- [Costa+, 2022] Costa, Emanuele, et al. "Deep-learning density functionals for gradient descent optimization." Physical Review E 106.4 (2022): 045309.

- [Medvidović+, 2025] Medvidović, Matija, et al. "Neural network distillation of orbital dependent density functional theory." Physical Review Research 7.2 (2025): 023113.


## Review Summary

The paper is well-written, and the main idea is clearly presented. The applicability of the proposed method to other domains is noteworthy. While the authors acknowledge the limitations of their theory for nonlinear functionals, addressing this would likely require a separate study, given the substantial challenges inherent in nonlinear functional analysis. Although the computational cost of the proposed model is higher than that of the baselines, it appears acceptable.

That said, there remain critical issues regarding the comparison with prior work, which may affect the central claims of the paper. For this reason, I am currently inclined to recommend rejection. However, I remain open to revisiting my evaluation and look forward to further discussion with the authors.

---

> ### Author Rebuttal · Authors · 2025-07-29
>
> Thank you for your detailed review and for the suggested related works. We discuss these and some additional points below:
>
> ### Related Works
> Thank you for providing such an extensive list of prior works. Many of these are new to us, so we appreciate the perspective and view they provide. Since the submission, we have also found other works that approximate functionals with an integral kernel [1,2], and other reviewers have also pointed out similar works [3, 4]. We arrived upon the idea independently, which unfortunately accounts for much of the language/framing used in the paper. We are happy to change this, especially since we have a novel perspective on using functionals in Hamiltonian mechanics to evolve PDEs. The related works will be expanded to include suggested prior works on approximating functionals and Section 3 will present intuition on their capabilities from a theoretical standpoint, rather than a derivation. Any language claiming to be the first work on approximating functionals will be changed. Beyond these general revisions, we also discuss some differences to specific works:
>
> #### Comparison to Miyagawa & Yokota, 2024
> Miyagawa and Yokota approximate scalar functionals by a SIREN that takes time and basis coefficients as input, and outputs a functional. In this form, it is only applicable to FDEs. A direct extension to PDEs without introducing Hamiltonian mechanics isn’t obvious, since functionals are usually not involved in solving PDEs. Ours (and prior works) also approximate functionals with an integral kernel to provide another perspective from Neural Operator literature and Riesz representations. Despite these differences, we are able to show that, with an appropriate choice of kernel function, the derivative of an integral kernel can be expressed as the cylindrical approximation of a functional derivative, which ties these two works together. The proof sketch is given above as a response to Reviewer snot.
>
> #### Comparison to Zhou et. al. and Naming of “Neural Functional”
> The similarities to this family of works are likely in just the name “Neural Functional” and the general idea of inputting functions to networks. The authors are mostly interested in principled approaches to learn with the weights of other neural networks, for tasks such as predicting test-time accuracy of models or classifying INRs based on their weights. In our work, we mainly consider functionals as a mathematical object that maps between functions and scalars, and the applications of functional derivatives in PDE prediction.
>
> However, we agree that the overlap in terminology is confusing and will work to remedy this. Specifically, we will change the title of the work to “Hamiltonian Neural PDE Solvers through Functional Approximation” (or something similar) to emphasize the novel perspective of Hamiltonian mechanics, which allows functional approximators to solve PDEs. Since the contribution of our work is no longer centered around introducing neural functionals (the mathematical object), we don’t see a need to introduce a new term and will refer to them more generically as functional approximators, integral kernels, learning the Hamiltonian, etc.
>
> #### Comparison to Machine Learning for DFT works
> These seem to be similar to the introduced method for approximating functionals, as they also employ a kernel integral (where the kernel is an energy density). Surprisingly, these works introduce kernel integration many years before they were developed as a theoretical foundation for neural operators. Beyond architecture, many works also make use of functional derivatives to approximate potentials for downstream use. We will make sure to mention these similarities, however our application of this paradigm to predict PDEs is still previously unexplored, due to the necessary introduction of Hamiltonian mechanics. Hamilton’s equations allow a mapping from scalar functionals to dynamical quantities of interest (du/dt) which differs from the usual Newtonian perspective in PDE works. Furthermore, learnable kernel integration in DFT arises since it resembles known equations and empirical functionals, however in PDEs, it is derived from operator approximation and Riesz representation theorems, which offers a perspective from functional analysis.
>
> ### Computing Functional Derivatives with AD
> Thank you for providing the relevant references. DFT is a field we are not familiar with, however, we recognize there are many similarities and will discuss them in our work. A similar line of work in molecular dynamics (MD) learns an energy functional and uses its spatial derivative (force) to advance a simulation in time. We made note of the MD domain in our related works, but will expand it to create a section about functional approximation/derivatives in other domains.
>
> ### Overstatements
> Thank you for pointing these out, we agree that the quoted lines are overstatements. We will remedy this by removing language that suggests that we introduce functional approximation, but rather use it as a tool to obtain neural PDE solvers that have better abilities to conserve Hamiltonians and extrapolate to new scenarios. Your intuition about the approach taken by Miyagawa & Yokota converging to ours is correct and is something we investigate in the above proof sketch and will discuss in the work.
>
> ### Clarifying Relationship to Other Neural Surrogates
> Thank you for the suggestion. We will expand on the benefits of our method (conserving energies, extrapolation) as well as some of the drawbacks (higher training/inference cost, more effort to derive Hamiltonian structure) with respect to common neural surrogates in the field.
>
> ### Statistical Significance
> Thank you for the suggestion. We have doubled the number of seeds (up to 6 seeds) in our experiments. Here is a comparison of one such table (Table 2 in the paper):
>
> **Table 1: Comparison of Mean/Std calculated across 3 or 6 seeds**
> | Model | Adv (3 seeds) | Adv (6 seeds) | KdV (3 seeds) | KdV (6 seeds) |
> |-------------------------|----------|-----------|-----------|-----------|
> | FNO | 0.81$\pm$0.17 |  0.83$\pm$0.17 |  69.5$\pm$8.2 | 68.0$\pm$10.5 |
> | Unet | 0.52$\pm$0.35 | 0.40$\pm$0.29 |  125.7$\pm$8.5 | 134.0$\pm$10.6 |
> | FNO (du/dt) | 0.044$\pm$0.002 | 0.048$\pm$0.007 |  75.5$\pm$1.9 | 77.6$\pm$3.6 |
> | Unet (du/dt) | 0.068$\pm$0.029 | 0.057$\pm$0.024 |  127.4$\pm$4.9 | 113.3$\pm$15.0 |
> | HNF | 0.0039$\pm$0.0002 | 0.0039$\pm$0.0008 |  150.9$\pm$3.3 | 151.1$\pm$3.0 |
>
> For brevity we do not present all the updated tables here. While some numbers have changed, the overall findings remain the same. We also add these error bars into Tables 1, 4, 5, 6 (across 6 seeds). Here is one such updated table (Table 1 in the paper):
>
> **Table 2: Results for Toy Examples, with Mean/Std calculated across 6 seeds**
> | | Base | | | OOD | | | Disc | | |
> |-------------------------|----------|-----------|-----------|-----------|-----------|-----------|-----------|-----------|-----------|
> | Metric | MLP | FNO | NF | MLP | FNO | NF | MLP | FNO | NF |
> | $\mathcal{F}_l[u]$ | 2e-05$\pm$2e-05 | 2.8e-04$\pm$1.6e-04 | 3e-16$\pm$2e-16 | 0.046$\pm$0.074| 0.27$\pm$0.24 | 2.13e-07$\pm$4.21e-07 | 49.29$\pm$11.05 | 49.70$\pm$11.16| 0.74$\pm$1.16|
> | $\delta\mathcal{F}_l/\delta u$ | 0.083$\pm$0.012| 0.066$\pm$0.034 | 0.0012$\pm$0.00053 | 0.11$\pm$0.026 | 0.12$\pm$0.077 | 0.0012$\pm$0.00053 | 2.64$\pm$0.029 | 2.70$\pm$0.073 | 0.077 $\pm$0.047 |
> | $\mathcal{F}_{nl}[u]$ | 0.029$\pm$0.025 | 0.016$\pm$0.016 | 0.002$\pm$0.0007| 6709$\pm$1567 | 6493$\pm$1614| 2908$\pm$816 | 381$\pm$256 | 322$\pm$266 | 55$\pm$50 |
> | $\delta\mathcal{F}_{nl}/\delta u$ | 1.325$\pm$0.388 | 2.097$\pm$1.902 | 0.045$\pm$0.041 | 1730$\pm$288 | 1723$\pm$295 | 1079$\pm$231 | 92$\pm$35 | 88$\pm$37 | 35$\pm$17 |
>
> For chaotic systems (KdV) or nonlinear functionals, spread can be large, which inflates standard deviations.
> ### Example about Future Applications
> In design optimization, for example, we are trying to find a set of design variables (i.e. points that define an airfoil spline) that minimize or maximize some scalar functional (i.e., drag of an airfoil). Most design variables are a discrete representation of an underlying object. For example, an airfoil may be defined by 100, 200, …, 1000 spline points, so a direct map from the coordinates to the functional is usually infeasible or at least not generalizable to different discretizations. Parameterizing the functional as a kernel integral is a natural remedy to this problem, and the gradient calculated by AD could be potentially used to adjust initial design variables towards a more desirable state.
>
> ### Adoption of FiLM
> We perform a series of ablation studies to justify the use of FiLM, in Table 5 in the response to Reviewer fiA5. We find that it can improve the performance of HNFs in some problems, but in general this is an empirical choice. Other options for incorporating conditional information into $k(x, u)$ can be investigated in the future.
>
> ### Extensions to FDEs
> This is an interesting direction, and it may be possible. Replacing F[u] with a learnable kernel integral seems possible, but one potential challenge will be using two quadratures to approximate dF[u]/dt: one for integrating the kernel and another for integrating the functional derivatives. An interesting aspect of our approach is it can be used independently of a basis or basis coefficients.
>
> ### Robustness to Grid Size
> We find that HNFs are approximately discretization invariant, shown in Tables 2-4 in our response in Reviewer fiA5. After training on a given resolution, they can be queried at coarser or finer grids without significant changes in performance.
>
> ### References
> 1. https://arxiv.org/abs/2408.01362
> 2. https://arxiv.org/abs/2402.06031
> 3. https://openreview.net/pdf?id=aQuqw6eVKP
> 4. https://arxiv.org/abs/2205.03017
>
> Thank you for the review. If there are any additional concerns/questions, we would be happy to discuss them.

---

> > ### Comment · Reviewer_a4TR · 2025-08-05
> >
> > Thank you very much for the detailed response, which addressed some of my concerns.
> > I would like to encourage the authors incorporate all discussions on additional related works in the revision.
> > I also appreciate the authors' tremendous effort in adding error bars and increasing the number of runs.
> > Although some results do not appear statistically significant, the variances of the proposed method are sometimes smaller than baselines, which can be beneficial for practitioners. Additionally, the results help clarify the conditions under which the proposed method works well or not.
> >
> > I have read other reviews and found that all reviewers also pointed out that the authors did not cite relevant prior work.
> > However, the authors effectively addressed the concern in the rebuttal: Although this paper has an overlap in its scope with several prior studies, the authors develop their method on the Hamiltonian formulation, which can be an independent contribution of this paper.
> > Additionally, other comments, concerns, suggestions from the reviewers were also addressed, including additional theoretical analysis and experimental results that make the paper's result more convincing.
> > I believe that the paper's quality has significantly improved through the discussions with the reviewers, and the paper's contributions have been much clarified.
> >
> >
> > ======================================================
> >
> > In summary, the paper is well-written and easy to follow, and the applicability of the proposed method to other domains is noteworthy, as noted in my review (see Review Summary). Additionally, through the rebuttal, the authors clarified their contributions in relation to prior studies that were not cited in the original manuscript.
> >
> > However, I am still concerned about the statistical significance of the experimental results because several error bars (signal-to-noise ratios) are large, and some of the results, if not all, are inconclusive, potentially affecting reproducibility of the work.
> > Also, while the authors effectively addressed many concerns raised by the reviewers, the paper requires a major revision due to the aforementioned lack of citations, as outlined in the responses for all reviewers.
> >
> > Overall, I think the contributions of the paper slightly outweigh the concerns, and I will raise my score from 3 to 4 (Borderline accept).
> > I kindly encourage the authors to run experiments as many times as possible to address the notorious variability of DNNs.
> > I would recommend using statistical tests where appropriate.
> > I have encountered many papers whose results I could not reproduce, even when they were published at top conferences.
> > Including error bars will not only strengthen the paper's credibility but also help protect the authors' reputation as researchers, in my humble opinion.
> >
> > Again, I appreciate the authors' hard work in responding to my questions and comments.

---

### Official Review · Reviewer_snot · 2025-06-28

**Clarity:** 3
**Significance:** 3
**Originality:** 4
**Rating:** 5
**Confidence:** 3

**Summary:**

This work introduces a novel neural network called a Neural Functional (NF). Rather than learning functions from inputs to outputs, it learns to map input functions to a single number — a functional. The authors apply this concept to construct a particular model called the Hamiltonian Neural Functional (HNF), which is capable of learning physical systems' energy-like values (Hamiltonians) that obey particular rules (PDEs in Hamilton form). Of particular importance is predicting how these systems evolve in time while keeping their total energy constant. The model exploits automatic differentiation to compute the evolution of the system. It is experimented with in simple examples, one-dimensional PDEs such as Advection and KdV equations, and a two-dimensional shallow water model. HNF is compared to various models such as MLPs, FNOs, and U-Nets in this work.

**Questions:**

### **Questions for Authors**
In addition to the weaknesses noted above, we ask the authors to consider the following questions:

1. Can you explain when and why your model can learn complicated (nonlinear) functionals? Do you have any theory to back this up?
2. Why should we believe that using autograd on a neural network gives the correct kind of derivative (the functional derivative) in your setting?
3. Your method is built around learning Hamiltonians and using them to evolve systems over time while conserving energy. However, many real-world physical systems are not purely Hamiltonian — they may include friction, damping, external forcing, or other types of energy gain/loss. Can your framework be adapted to handle these types of systems? If not, how much does this limit the range of problems your method can be applied to in practice? It would be helpful to understand whether your approach can be extended to include non-conservative dynamics, or if it's restricted to idealized, energy-conserving cases only.
4. How does the choice of grid or resolution affect your results? Have you tested what happens when the inputs are more coarsely sampled?
5. Why didn’t you compare your model to other physics-based methods like DeepONet, PINA, or Symplectic Neural Operators?
6. What is the actual runtime, memory cost and training data demand of your model during prediction compared to FNO or U-Net, especially on the 2D shallow water test?
7. How do we know how much training data is enough?
8. How does one represent the impacts of distributed inputs like bed topography or diffusion coefficient in the equations?

I'm putting Borderline Accept for now due to the novel idea and giving the authors the benefit of the doubt, but if the questions got no good answers I reserve the right to mark it down.

**Ethical Concerns:**

["NO or VERY MINOR ethics concerns only"]

**Final Justification:**

Questions have been addressed thoroughly. Idea is novel. Results look promising.

**Quality:**

3

**Strengths And Weaknesses:**

### **Strengths**
- The paper explores a fresh idea — learning energy-like quantities (called Hamiltonians) directly as functionals, which is different from how most models are designed.
- The authors try to base their method on solid math (using something called the Riesz theorem), which helps explain how it works, at least for simple cases.
- The experiments show that the proposed model (HNF) does a good job keeping the system’s energy stable over time, which is important for long simulations.
- The authors test their model on various examples — simple mathematical expressions, one-dimensional equations, and a more complex two-dimensional fluid system, which gives a broad look at how well the method works.

### **Weaknesses**
- The theory the paper relies on (Riesz theorem) only works for simple cases, but the authors use it for complex, nonlinear situations without clear justification.
- There’s no formal proof that the model works well or converges, not even for simple cases.
- Some similar past work is not discussed or compared to, including other models that also learn energy functions or solve physics-based problems.
- In some tests, like the 2D shallow water case, the model keeps energy well, but doesn’t predict the actual system behavior better than others like U-Net. Improvements or explanations are needed.
- The model may be slow during prediction because it needs backward passes every step, but this isn’t clearly shown or discussed in the main results.
- The method requires the knowledge of the J operator, which was said to be trivial in the paper but may not be that trivial if we learn from real data and the real system is unclear. It also requires the preparation of derivative as training data `∂u/∂t` for training. The computing cost of this was not discussed.
- The impact of grid resolution (training and test accuracy, and compute time) was not discussed. For high-resolution simulations, this approximation may not be trivial. And we need to ensure the cost is not higher than the direct numerical simulation.
- The authors did not discuss if their system works for problems with parameters, i.e., shallow water equation with bed topography, or advection-diffusion equation with varying diffusion coefficient.

---

> ### Author Rebuttal · Authors · 2025-07-29
>
> Thank you for the review. Apologies for the brevity of tone, since there were many suggestions and questions:
>
> ### Additional Theory
> With the feedback of Reviewer a4TR, we have found additional insight that connects our approach with an approximation of functional derivatives.Consider functionals $F[u]: H \rightarrow \mathbb{R}$, where functions $u(x) \in H$ are in a Hilbert space $H$ with inner product $\langle\cdot, \cdot\rangle_H$. $u(x)$ can be written as $u(x) = \sum_{k=1}^\infty \langle u,\phi_k\rangle\phi_k(x)$, with a basis $\{\phi_1, \phi_2, \ldots\}$. Truncating the sum to a finite number of terms $m$ allows the cylindrical approximation of functionals [1]:
>
> $$
> 	f(a_1, a_2, \ldots, a_{m}) := F[\sum_{k=1}^{m}a_k\phi_k(x))]
> $$
>
> where $a_k$ are basis coefficients $\langle u,\phi_k\rangle$. With the cylindrical approximation, functional derivatives can be written as:
>
> $$
> 	\frac{\delta F}{\delta u} \approx \sum_{k=1}^{m} \frac{\partial f}{\partial a_k}\phi_k(x)
> $$
>
> which converges to the true derivative as $m\rightarrow\infty$. To discretize the derivative, we define a set of $n$ points $\mathbf{x} = [x_1, x_2, \ldots, x_{n}]$, which results in the discrete approximation:
>
> $$
> 	\sum_{k=1}^{m} \frac{\partial f}{\partial a_k}\phi_k(x) \approx \sum_{k=1}^{m} \frac{\partial f}{\partial a_k}\phi_k(\mathbf{x}) = \begin{bmatrix}
>         	\sum \frac{\partial f}{\partial a_k}\phi_k(x_1) \\\\
>        	\sum\frac{\partial f}{\partial a_k}\phi_k(x_2)  \\\\
>        	\vdots \\\\
>        	\sum\frac{\partial f}{\partial a_k}\phi_k(x_{n})
>      	\end{bmatrix}
> $$
>
> We are interested if the gradient of a kernel integral can represent this discrete object. We skip to the final result, which uses kernel functions of the form $\kappa_\theta(x, \mathbf{u})$, where the kernel depends locally on the coordinate $x$ and globally on function values $\mathbf{u} = [u_1, u_2, \ldots, u_n]$.
>
> The gradient of a nonlinear, global kernel integral can be written as:
>
> $$
> 	\nabla_u \int_\Omega \kappa_\theta(x, \mathbf{u})u(x) \text{d}x \approx \mu \nabla_u(\sum_{i=1}^{n} \kappa_\theta(x_i, \mathbf{u})u_i) = \mu\begin{bmatrix}
>     	\frac{\partial}{\partial u_1} (\sum\kappa_\theta(x_i, \mathbf{u})u_i)\\\\
>    	\frac{\partial}{\partial u_2} (\sum\kappa_\theta(x_i, \mathbf{u})u_i)\\\\
>    	\vdots \\\\
>    	\frac{\partial}{\partial u_{n}} (\sum\kappa_\theta(x_i, \mathbf{u})u_i)
>  	\end{bmatrix}
> $$
>
> with quadrature weight $\mu$. The expression $\frac{\partial}{\partial u_{j}}(\sum\kappa_\theta(x_i, \mathbf{u})u_i)$ can be expanded:
>
> $$
> 	\frac{\partial}{\partial u_{j}}(\sum_{i=1}^n\kappa_\theta(x_i, \mathbf{u})u_i) =  u_1\frac{\partial}{\partial u_{j}}\kappa_\theta(x_1, \mathbf{u}) + \ldots +  u_j\frac{\partial}{\partial u_{j}}\kappa_\theta(x_j, \mathbf{u}) + \kappa_\theta(x_j, \mathbf{u}) + \ldots +  u_n\frac{\partial}{\partial u_{j}}\kappa_\theta(x_n, \mathbf{u})
> $$
>
> In general, we can construct a kernel $\kappa_\theta$ such that $\frac{\partial}{\partial u_{j}}\kappa_\theta(x_i, \mathbf{u})$ is different for $i=\{1, \ldots, n\}$. An example is provided based on FiLM:
>
> $$
> 	\kappa_\theta(x_i, \mathbf{u}) = [W^u\mathbf{u}]^i(W^xx_i +b^x) + [b^u]^i
> $$
>
> where $W^u \in \mathbb{R}^{n\times n}, W^x \in \mathbb{R}^{1x1}, b^u \in \mathbb{R}^{n},b^x \in \mathbb{R}^1$ and $ [\cdot]^i$ is an operation that takes the $i$-th row of a column vector. In this case, $\frac{\partial}{\partial u_{j}}\kappa_\theta(x_i, \mathbf{u})$ is equal to a function $d_{ij}(x_i) = W^q_{ij}W^xx_i$. With deeper kernels, functions $d_{ij}$ can become more expressive, and we can also absorb the basis coefficients into $d_{ij}$.
>
> The $j$-th row of the gradient vector can be written as a sum of functions $\sum_{i=1}^{n} d_{ij}(x_i)$, which reduces to the cylindrical approximation when letting $d_{ij}$ be arbitrary. Letting $n \rightarrow \infty$ increases the number of functions $d_{ij}$ and coordinates $x_i$ to increase the number of bases to approach the true derivative as well as convert from a discrete to continuous representation. The cylindrical approximation makes no assumption of linearity, therefore, using a global, nonlinear kernel integral can represent a functional derivative. The kernel function must have enough inputs; if it restricted to a linear kernel $\kappa(x)$ or a local kernel $\kappa(x, u(x))$, the same calculation only yields a single basis function. Intuitively, the kernel needs to depend on enough inputs so its gradient can be expressive enough.
>
> ### Discussion of Prior Works
> Our related works include a few works from molecular dynamics (MD) that learn energy functionals and use their derivatives to update the simulation. Reviewer a4TR also suggested many works from the DFT field that use functional approximation, which we will include.
>
> ### Accuracy vs. Conservation of Energy
> In general, adding inductive biases to conserve energy can help accuracy, but is not strictly true. For example, learning the identity map conserves energy (since the state does not evolve), but is not accurate. While HNFs are not better than the Unet, the additional inductive biases help it to extrapolate to unseen conditions, which is the axis upon which we seek to improve.
>
> ### Prediction Speeds
> We discuss this in Section 5 and in Appendix B.2 of the paper and in Table 8 of our response to Reviewer fiA5, where we run timing experiments of different models and numerical solvers. The overhead of backprop increases the computational cost with respect to other models and scales with the number of query points, which, as you mention, is a limitation.
>
> ### J Operator and du/dt Preparation
> On real-world systems, prior works sometimes make an ansatz for contributions to the dynamics that conserve energies or satisfy certain properties (for example in climate [2]). If the Hamiltonian structure is not known, we can make a guess for its form based on similar, identifiable systems, and model other dynamics separately. In general, preparing du/dt for training has negligible overhead. It uses finite differences, which is a constant number of operations per sample, and du/dt can be precomputed and cached for training as well.
>
> ### Impact of Grid Resolution
> We perform a set of studies to compare the accuracy of models trained on a given resolution and queried at coarser or finer resolutions, in Tables 2-4 in our response to Reviewer fiA5. At test-time, HNFs can be used at different grids and resolutions without significant changes in accuracy. As for the computational cost, we compare the cost of our model to numerical solvers in Table 8 of our response to Reviewer fiA5. While the cost of our model does scale with the input resolution, the cost of numerical solvers also scales with increasing resolution.
>
> ### Effect of Coefficients or Sources/Sinks
> It is possible to use our work with varying coefficients and source terms with changes to the Hamiltonian structure. We provide examples below by introducing coefficients to 1D Advection as well as varying topography to 2D SWE.
>
> In the presence of coefficients, the Hamiltonian for 1D Advection is:
>
> $$
> 	\mathcal{H}[u] = \int_\Omega -\frac{c}{2}(u(x))^2 dx
> $$
>
> which recovers the equation $-c\frac{\partial u}{\partial x} = \frac{\text{d}u}{\text{d}t}$ To accommodate varying coefficients, we can train HNFs by inputting coefficients to the model: $\mathcal{H}_\theta(u, x, c)$. In cases where the coefficients can be factored out (i.e., $\frac{\delta \mathcal{H}[u, c]}{\delta u} = c\frac{\delta \mathcal{H}[u]}{\delta u}$), once can simply scale the model output $\nabla_u\mathcal{H}(u, x)$ by $c$.
>
> Spatially-varying beds $b(x, y)$ can accelerate or decelerate flows, which acts as a source term on the momentum balance. The shallow water equations become:
>
> $$
> 	\partial_th + \nabla\cdot (\mathbf{v}h) = 0, \qquad \partial_t\mathbf{v} + \mathbf{v}\cdot\nabla\mathbf{v}=-g\nabla h -gh \nabla b
> $$
>
> $h(x,y)$ now represents the height above $b(x,y)$. This source term changes the Hamiltonian to:
>
> $$
> 	\mathcal{H}^b[\mathbf{u}] = \mathcal{H}^b[v_x, v_y,h] = \int_\Omega \frac{1}{2}h(v_x^2 + v_y^2 ) + \frac{1}{2}gh^2(1+b) \text{d}A
> $$
>
> The operator matrix $\mathcal{J}$ is left unchanged. One can check that applying Hamilton's equations recovers the modified shallow-water equations. When comparing this derivation to one with constant bathymetry, one can show that:
>
> $$
> 	\frac{\delta \mathcal{H}^b[u]}{\delta u} = \frac{\delta\mathcal{H}[u]}{\delta u} + \begin{bmatrix}
>    	0\\\\
>    	0\\\\
>    	ghb
>  	\end{bmatrix}
> $$
>
> Therefore, source terms can be potentially captured by adding terms to model outputs. In PDEs where a decomposition like this is not possible, the source term will need to be added to the model input to produce the output $\nabla_u\mathcal{H}_\theta(u, x, b)$.
>
> ### Extensions to Non-Conservative Systems
> Similar to [4], we may be able to model the dissipative dynamics jointly with the Hamiltonian with the Dissipation $[D(u, x), H(u, x)]$. For continuum systems, the theory governing this is not well-understood. We can potentially make an ansatz about how the derivative of the dissipation relates to the dynamics $du/dt$ but this is not well-studied.
>
> ### Comparison to Other Models
> We run more baselines for 2D SWE, in Table 1 of our response to Reviewer fiA5, including a physics-informed baseline.
>
> ### Training Data
> To determine how much training data is needed, we train HNFs on datasets downsampled by a factor f = ½, ¼ or ⅛ . In general, increasing the dataset size consistently improves model performance.
>
> **Table 1: HNF Performance vs. Training Set Size**
> | Dataset Size | Adv | KdV | SWE-Sines |SWE-Gaussian|
> |-------------------------|----------|-----------|-----------|-----------|
> | f=1/8 | 1.320 | 71.00 | 0.0293 | 0.0243
> | f=1/4 | 0.302 | 71.25 | 0.0274 | 0.0230 |
> | f=1/2 | 0.075 | 95.25 | 0.0254 | 0.0227 |
> | f=1 | 0.003 | 151.75 | 0.0249 | 0.0216 |
>
> ### References
> 1. https://arxiv.org/abs/2410.18153
> 2. https://arxiv.org/abs/2404.10024
> 3. https://arxiv.org/abs/2201.10085

---

> > ### Comment · Reviewer_snot · 2025-08-04
> >
> > I thank the authors for responding to my comments thoroughly, and the new experiments used to answer my (and other reviewers') questions. I will raise my score accordingly.

---

### Official Review · Reviewer_fiA5 · 2025-06-28

**Clarity:** 4
**Significance:** 3
**Originality:** 3
**Rating:** 4
**Confidence:** 4

**Summary:**

This paper proposes a new architecture for learning neural functionals inspired by the Riesz representation theorem and applies this architecture to neural surrogate modeling of Hamiltonian PDE systems. The authors show that their method outperforms baseline models such as FNO and U-Net on a variety of problems, including 1d and 2d PDEs.

**Questions:**

1. Did the authors perform ablations to justify their choice of the SIREN architecture?
2. To better understand the timescale of solving these PDEs, how fast does it take numerical solvers to generate the same amount of data as in the inference step (Table 6)?

**Ethical Concerns:**

["NO or VERY MINOR ethics concerns only"]

**Final Justification:**

My initial concerns were an expanded comparison with prior works, a comparison with more baseline methods, and experiments in super-resolution and changing grids at test time. The authors addressed my points thoroughly by providing experiments on baselines, discretization invariance, and numerical solver timing. For this reason, I decided to increase my score from my initial response.

**Limitations:**

The authors discuss the limitations of their work in Section 5.

**Quality:**

3

**Strengths And Weaknesses:**

**Strengths:**
The paper is well-written and well-motivated. The background theory is explained clearly for all audience members. The experiments follow a logical pattern, going from toy examples to 1d problems to 2d problems. The evaluation criteria goes beyond simple MSE and includes plots of the Hamiltonian over time.

The authors write that to the best of their knowledge, “this is the first time an architecture has been developed to learn function to scalar mappings.” I am only aware of one other prior work that has done this. The discriminator module in [1] uses a “neural functional layer” to map a function to a real number.

Despite some overlap between this work and [1], the authors clearly put effort into identifying the important architectural considerations for the neural functional, and the application to Hamiltonian systems is novel to my knowledge.

**Weaknesses:**
The authors should discuss similarities and differences between their method and the “neural functional” introduced in [1].

While the experiments are thorough and detailed, the number of baselines could be expanded, particularly for the 2d shallow water equations where the difference in error between HNF, FNO, and U-Net is less than for the other problems. There are a variety of architectures that have been introduced in recent years that have been competitive with (or outperformed) FNO and U-Net in some problems.

Since HNF is agnostic to the resolution, I would suggest adding experiments using different resolutions and grids. For instance, zero-shot super-resolution can be compared with FNO and other neural operators. Training on multiple grids or resolutions can also be performed. Out-of-domain generalization experiments to different grids would be interesting as well.

**Minor notes:**
1. Typo “its functional derivative can obtained” in line 38.
2. Typo “IN” in line 144.
3. In Figures 2 and 3, the red and pink lines may appear too similar for some readers.

**References:**
[1] “Generative Adversarial Neural Operators”

---

> ### Author Rebuttal · Authors · 2025-07-28
>
> Thank you for taking the time to write a review and for your suggestions. We have implemented many of these below:
>
> ### Related Works
> Thank you for providing the reference. Since submitting the paper, we have also found more works that introduce a neural functional parameterized by a kernel integral [1, 2] as well as related works from other reviewers [3, 4] that are similar. Since we arrived upon the idea independently, much of the language in our paper unfortunately suggests that it is a novel idea. We will work to remedy this by changing the title and framing of the work, as well as amending claims of discovering this.
>
> Despite the existence of prior works, there are some innovations that we make from independently developing functionals for Hamiltonian systems. Firstly, we make an effort to verify that kernel integrals can approximate functional derivatives with analytically understood systems. Secondly, we use neural functionals to do a PDE prediction task, which is enabled by Hamiltonian mechanics. Lastly, we investigate implementing the kernel as a conditioned neural field with more advanced architectures (FiLM/SIREN), which is empirically necessary for good performance (see ablation studies below). Specific to the referenced work from Rahman et. al., our approach introduces more complexity into the kernel, since they only work with a linear, local kernel parameterized by an MLP, and the referenced work does not make use of functional derivatives.
>
> ### Baselines
> We have included two additional baselines for 2D SWE, a Transolver [5] and PINO [6] baseline. We decided to compare against Transolver, since it introduces a new architecture (attention-based) and has shown empirically good performance, and PINO, to compare against an established, physics-informed method. The modified results for 2D SWE are:
>
> **Table 1: Rollout Error of Models on 2D SWE**
> | Model             	| Sines | Pulse |
> |-------------------------|----------|-----------|
> | Transolver (3.6M) | 0.084$\pm$0.0081 | 0.122$\pm$0.0074 |
> | FNO (7M) | 0.057$\pm$0.002 | 0.117$\pm$0.0009 |
> | PINO (7M) | 0.053$\pm$0.005 | 0.114$\pm$0.0003 |
> | Unet (3M) | 0.010$\pm$0.0014 | 0.042$\pm$0.0006 |
> | HNF (3M) | 0.026$\pm$0.0003 | 0.021$\pm$0.0015 |
>
> We find that adding a physics-informed loss to FNO (i.e., PINO), helps to improve the performance slightly. The loss formulation we employed was: $L = L_{data} + w*L_{PDE}$, where $L_{PDE}$ is the residual of the 2D SWE equations, calculated with finite differences. For a fair comparison, we do not perform test-time finetuning of PINO. We report results with $w = 0.0001$, which was the best-performing model after a hyperparameter sweep.
>
> Surprisingly, Transolver does not perform as well as other methods. We investigated this through an extensive hyperparameter sweep and different model sizes, but was unable to improve its performance beyond what is reported. As a whole, the results are consistent with prior works that suggest Unets, while simple, are a well-performing architecture [7] on regular grid problems (0.01 relative L2 error is also quite small, and is qualitatively indistinguishable from the ground truth).
>
> ### Discretization Invariance
> As suggested, we experiment with querying models at different discretizations than its training resolution. The results are below, with the training resolution in bold. Rollout errors are averaged over the validation set, where validation labels are either downsampled or re-solved at a higher resolution using the same initial conditions. Additionally, for 1D Advection, we run an experiment where the model is queried on an unseen grid (x=[0, 32], n_x=256), after being trained on a grid (x=[0, 16], n_x=128), which requires generating new validation samples.
>
> **Table 2: Rollout Error of Models at Different Resolutions, 1D Adv**
> | Resolution ($n_x$) | HNF | FNO (du_dt) | Unet (du_dt)
> |--------------|--------------|--------------|--------------|
> | $64$ | 0.0029 | 0.0639 | 1.1882 |
> | $\mathbf{128}$ | 0.0033 | 0.0639 | 0.0753 |
> | $256$ | 0.0055 | 0.0497 | 1.3177 |
> | $256, x=[0, 32]$ | 0.0021 | 1.3438 | 0.0723 |
>
> **Table 3: Rollout Error of Models at Different Resolutions, 2D SWE (Sines)**
> | Resolution ($n_x,n_y$) | HNF | FNO | Unet |
> |--------------|--------------| --------------| --------------|
> | $(64, 64)$ | 0.028 | 0.066 | 0.098 |
> | $\mathbf{(128, 128)}$ | 0.029  | 0.066 | 0.008 |
> | $(256, 256)$ | 0.026 | 0.066 | 0.110 |
>
> **Table 4: Rollout Error at Different Resolutions, 2D SWE (Pulse)**
> | Resolution ($n_x,n_y$) | HNF | FNO | Unet |
> |--------------|--------------|--------------|--------------|
> | $(64, 64)$ | 0.035 | 0.173 | 0.173 | 0.145 |
> | $\mathbf{(128, 128)}$ | 0.024 | 0.174 | 0.117 |
> | $(256, 256)$ | 0.019 | 0.174 | 0.225 |
>
> In all cases, HNFs are capable of zero-shot super-resolution, as is possible with Neural Operators. Additionally, the error is approximately constant across discretizations and on a new, unseen grid for 1D Advection. Predictably, FNO models have nearly constant error across discretizations. Interestingly, for an extrapolated grid in 1D Advection, the model struggles to make predictions. For Unet models, the error increases when the grid spacing changes, however for a constant spacing $dx$, it is able to extrapolate to unseen grids (i.e. [0, 16] -> [0, 32] with $dx$ unchanged).
>
> ### Ablations
> We perform ablations to study the effect of different model choices on HNF performance, across all PDEs (1D Adv, 1D KdV, 2D SWE-Sines). We consider the effect of different kernels (linear/nonlinear), architectures (MLP/SIREN), conditioning mechanisms (concat/FiLM) and receptive fields (local/global). Since this design space is large, we consider the effects of incrementally adding model features, where the base model is specified as (Linear, MLP, concat, local), and each successful model is cumulative:
>
> **Table 5: Ablation Studies for Different PDEs**
> | Model | 1D Adv | 1D KdV | 2D SWE-Sines |
> |-------------------------|----------|-----------|-----------|
> |      | Rollout Err. | Corr. Time | Rollout Err.|
> | Base | 1.248 | 51.75 | 0.113 |
> | +Nonlinear | 0.028 | 73.00 | 0.034 |
> | +FiLM | 0.016 | 65.25 | 0.030 |
> | +SIREN | 0.003 | 73.25 | 0.026 |
> | +Global | 0.057 | 151.75 | 0.024 |
>
> In general, using a SIREN vs. MLP architecture improves performance. We can see that using a nonlinear kernel is also important, as all Hamiltonians tested are nonlinear. In some cases, design choices such as using a global kernel or FiLM conditioning are also important.
>
> ### Numerical Solver Timing
> We update Table 6 with the runtime of a single step of numerical solvers, compared to neural solvers. Numerical solvers are run on a AMD Ryzen Threadripper PRO 5975WX 32-Core CPU due to lack of GPU compatibility. In the case of 1D Advection, the analytical solution is used. Furthermore, randomly sampled initial conditions can cause variability in the stiffness of the solution, therefore, solver runtimes are averaged across 10 samples. We note that the 2D SWE solver (PyClaw) uses a Fortran compiler and is optimized for hyperbolic PDEs, while the KdV solver is purely Python-based. Despite the additional overhead of backprop in HNFs, it is still modestly faster than a dedicated numerical solver. For a fairer comparison, the numerical solver would need to be coarsened to match the accuracy of the neural solvers (which are pretty accurate), however, this still serves as a point of reference and gives an understanding of the general timescales involved.
>
> **Table 8: Time/Step, in ms**
> | Model | Adv | KdV | SWE-Sines |
> |-------------------------|----------|-----------|-----------|
> | FNO | 0.083 | 0.091 |  0.967 |
> | Unet | 0.138  | 0.146 |  1.345 |
> | HNF | 0.126 | 0.228 |  4.547 |
> | Numerical Solver| 0.000 | 110.9 |  33.26 |
>
> ### References
> 1. J. Bunker et. al. Autoencoders in Function Space, https://arxiv.org/abs/2408.01362
> 2. D. Huang et. al. An operator learning perspective on parameter-to-observable maps, https://arxiv.org/abs/2402.06031
> 3. Z. Hu et. al. Neural Integral Functionals, https://openreview.net/pdf?id=aQuqw6eVKP
> 4. T. Miyagawa et. al. Physics-informed Neural Networks for Functional Differential Equations: Cylindrical Approximation and Its Convergence Guarantees, https://arxiv.org/abs/2410.18153
> 5. H. Wu et. al. Transolver: A Fast Transformer Solver for PDEs on General Geometries, https://arxiv.org/abs/2402.02366
> 6. Z. Li et. al. Physics-Informed Neural Operator for Learning Partial Differential Equations, https://arxiv.org/abs/2111.03794
> 7. G. Kohl et. al. Benchmarking Autoregressive Conditional Diffusion Models for Turbulent Flow Simulation, https://arxiv.org/abs/2309.01745v3
>
> If there are any further questions/concerns, we would be happy to address them. Thanks!

---

> > ### Comment · Reviewer_fiA5 · 2025-08-04
> >
> > I thank the authors for their hard work in responding to my questions and comments. I have increased my score accordingly.

---

### Official Review · Reviewer_qFJp · 2025-07-03

**Clarity:** 3
**Significance:** 3
**Originality:** 3
**Rating:** 5
**Confidence:** 4

**Summary:**

This paper introduces a neural network ansatz for learning the Hamiltonian functional, which can be used to model dynamics governed by a PDE. The paper parameterizes the functional as a local integral functional, whose functional derivatives can be obtained via AD. The approach is evaluated on 1D advection, KdV, and 2D shallow water equations, demonstrating improved energy conservation and stability compared to existing neural PDE surrogates.

**Questions:**

- In line 101, you mention that there exists $f(u)$ that can serve as a Riesz-like representation for a nonlinear functional. However, I don't see that there is any guarantee that $f$ can be approximated with $\kappa_{\theta}(x, u(x))$. For example, if $H[u]$ is non-local, then $\kappa_{\theta}(x, u(x))$ can never approximate $f$ since it only depends on $u$ pointwise.
- Why is it that in algorithm 1, you train on the functional derivative, while in the toy example, you train on the kernel? It is confusing that you also mentioned that fitting to the kernel also yields the correct functional derivative for the toy problem. If that's true, why change it for the actual problems?

**Ethical Concerns:**

["NO or VERY MINOR ethics concerns only"]

**Final Justification:**

I will maintain my initial score, since the author has resolved my concerns and questions during rebuttal, and the rebuttal with other reviewers did not bring up new information that would overrule my initial assessment.

**Limitations:**

yes

**Quality:**

3

**Strengths And Weaknesses:**

Strengths
- The mathematical background on Hamiltonian mechanics is appropriately introduced so that the paper is self-contained
- The author tested the OOD performance of the model
- Compared to previous methods, the proposed method captures the PDE dynamics in a more principled way
- Demonstrates practical benefits, including improved stability and energy conservation, especially for long rollouts

Weaknesses
There are existing works (e.g., https://openreview.net/pdf?id=aQuqw6eVKP) that have essentially the same setup, albeit even more general (semi-local functional, whereas in this paper the functional is local). The novelty of the paper comes from applying neural functionals to the specific setting of Hamiltonian mechanics. I feel that the author should clearly state the paper's contribution both in the title and the abstract. The keyword Hamiltonian should be mentioned in the title. Personally, I had very different expectations before reading the paper, by looking on the title.
- Using quadrature to approximate the integral makes this method not scalable in the domain dimension
- The reason that the functional derivative here can be directly be computed directly with AD could be explained in a much clearer way, for example, using the Euler-Lagrange formula
- Typo on line 144

---

> ### Author Rebuttal · Authors · 2025-07-28
>
> Thank you for taking the time to review our work and provide feedback. Please find a response to your questions/concerns below:
> ### Related Works and Framing
> Thank you for providing the existing work. Since the submission, we have also found other works that propose functionals approximated by an integral kernel [1,2], and other reviewers have also pointed out similar works [3, 4]. We arrived upon the idea independently, which unfortunately accounts for much of the language/framing used in the title and paper. We are happy to change this, especially since we have a novel perspective on using neural functionals in Hamiltonian mechanics to evolve PDEs. Indeed, this motivates much of our paper, such as verifying the validity of functional derivatives on toy problems and investigating better architectures (SIREN, FiLM). The title will be changed to: “Hamiltonian Neural Solvers through Functional Approximation” (or something similar), and the paper will be revised accordingly. In particular, the related works will be expanded to include prior work on integral kernel functionals and Section 3 will present intuition on their capabilities from a theoretical standpoint, rather than a derivation. Any language suggesting to be the first work on approximating functionals will be changed.
> ### Quadratures and Scalability
> This is a limitation, especially in 3D where numerical integration is very expensive. One potential avenue to alleviate this is to train on low-resolution data and only use the desired resolution during inference. We find that HNFs can be discretization invariant, which we verify in experiments suggested by Reviewer fiA5.
> ### Euler Lagrange formula
> Thank you for bringing this to our attention, it is a very interesting perspective. We will discuss the relationship of the Euler Lagrange equation to functional derivatives, as well as go through some examples, such as with the 1D Advection equation.
> ### Approximation Guarantees
> You are correct that an assumption of compact support would be necessary to approximate a functional with an integral kernel. In PDE problems this is usually a valid assumption, as most practitioners are interested in a physical response in a predetermined domain. Beyond compact support, we will make it clear that more assumptions are needed about H[u], as it must exist in the dual space of functions that are: (1) continuous, (2) have a uniformly continuous derivative, (3) have compact support, (4) have bounded derivatives. These are given in Appendix C.1. Similar restrictions also exist for Neural Operators in order to achieve their approximation guarantees.
> ### Training Differences
> This is a great question, and one we will clarify in the work. When constructing the toy problems, we tried to use Algorithm 1 to train the FNO/MLP models, however they did not succeed presumably since their gradients calculated by AD don’t allow them to learn underlying functionals. Therefore, to make things fair in the toy problems, we trained all models to fit the scalar-valued functionals, which has the added benefit of evaluating if models can implicitly learn functional derivatives. For the rest of the work, we also tried fitting the scalar-valued Hamiltonians, however, this had an unintended consequence. Since the Hamiltonian is conserved over time, models would usually learn the identity map for its functional derivative, resulting in the PDE not evolving in time (but still conserving the Hamiltonian). To allow the model to learn dynamical information, fitting it to the functional derivative was necessary.
>
> ### References
> 1. J. Bunker et. al. Autoencoders in Function Space, https://arxiv.org/abs/2408.01362
> 2. D. Huang et. al. An operator learning perspective on parameter-to-observable maps, https://arxiv.org/abs/2402.06031
> 3. M. Rahman et. al. Generative Adversarial Neural Operators, https://arxiv.org/abs/2205.03017
> 4. T. Miyagawa et. al. Physics-informed Neural Networks for Functional Differential Equations: Cylindrical Approximation and Its Convergence Guarantees, https://arxiv.org/abs/2410.18153
>
> Looking forward to any future discussions if you have any questions/concerns.

---

> > ### Comment · Reviewer_qFJp · 2025-08-08
> >
> > Thank you for your detailed reply. My initial concerns and questions are well addressed. I have also read your discussion with the other reviewer. I look forward to seeing your final draft, where all the discussed items are incorporated.

---

### Decision · Program_Chairs · 2025-09-17

**Decision:**

Accept (poster)

**Comment:**

This paper proposes a novel neural architecture grounded in functional analysis, combining the Riesz representation theorem, integral kernel operators, and neural fields to represent linear (and potentially nonlinear) functionals. A tailored training algorithm for functional Hamiltonian problems is also introduced. The technical contribution is significant, as it demonstrates how neural networks can be leveraged to simplify the treatment of complex functionals, which has been a longstanding challenge in numerical analysis.
The reviewers consistently praised the clarity of the paper. The exposition is well-structured, with sufficient background provided on Hamiltonian mechanics, functional analysis, and neural fields, making the paper accessible to a broad audience. The work also shows strong reproducibility, with code made available. Experimentally, the model achieves superior performance over established baselines on several PDE tasks, highlighting its effectiveness, though at a somewhat higher computational cost.
Beyond technical merit, the proposed approach has promising broader impacts. Its generality suggests applicability to diverse domains such as density functional theory and functional differential equations. Overall, the combination of novelty, rigor, clarity, and potential for broad influence outweighs the computational overhead.